# Endoparasitoid lifestyle promotes endogenization and domestication of dsDNA viruses

**Benjamin Guinet**[1]*, **David Lepetit**[1], **Sylvain Charlat**[1], **Peter N Buhl**[2], **David G Notton**[3], **Astrid Cruaud**[4], **Jean-Yves Rasplus**[4], **Julia Stigenberg**[5], **Damien M de Vienne**[1], **Bastien Boussau**[1], **Julien Varaldi**[1]*

[1]Université Lyon 1, CNRS, Laboratoire de Biométrie et Biologie Evolutive UMR 5558, F-69622, Villeurbanne, France; [2]Zoological Museum, Department of Entomology, University of Copenhagen, Universitetsparken, Copenhagen, Denmark; [3]Natural Sciences Department, National Museums Collection Centre, Edinburgh, United Kingdom; [4]INRAE, UMR 1062 CBGP, 755 avenue 11 du campus Agropolis CS 30016, 34988, Montferrier-sur-Lez, France; [5]Department of Zoology, Swedish Museum of Natural History, Stockholm, Sweden

*For correspondence:
benjamin.guinet95@gmail.com
(BG);
julien.varaldi@univ-lyon1.fr (JV)

**Competing interest:** The authors declare that no competing interests exist.

**Abstract** The accidental endogenization of viral elements within eukaryotic genomes can occasionally provide significant evolutionary benefits, giving rise to their long-term retention, that is, to viral domestication. For instance, in some endoparasitoid wasps (whose immature stages develop inside their hosts), the membrane-fusion property of double-stranded DNA viruses have been repeatedly domesticated following ancestral endogenizations. The endogenized genes provide female wasps with a delivery tool to inject virulence factors that are essential to the developmental success of their offspring. Because all known cases of viral domestication involve endoparasitic wasps, we hypothesized that this lifestyle, relying on a close interaction between individuals, may have promoted the endogenization and domestication of viruses. By analyzing the composition of 124 Hymenoptera genomes, spread over the diversity of this clade and including free-living, ecto, and endoparasitoid species, we tested this hypothesis. Our analysis first revealed that double-stranded DNA viruses, in comparison with other viral genomic structures (ssDNA, dsRNA, ssRNA), are more often endogenized and domesticated (that is, retained by selection) than expected from their estimated abundance in insect viral communities. Second, our analysis indicates that the rate at which dsDNA viruses are endogenized is higher in endoparasitoids than in ectoparasitoids or free-living hymenopterans, which also translates into more frequent events of domestication. Hence, these results are consistent with the hypothesis that the endoparasitoid lifestyle has facilitated the endogenization of dsDNA viruses, in turn, increasing the opportunities of domestications that now play a central role in the biology of many endoparasitoid lineages.

## Editor's evaluation

This important manuscript employs a rigorous and multi-pronged comparative genomics approach to unravel how lifestyle modulates the acquisition and domestication of viral genetic elements in the genomes of hymenopteran insects. Using an extensive dataset of over 120 hymenopteran genomes, the authors provide convincing evidence that endoparasitism (where parasite development occurs within hosts) facilitates the uptake and domestication of double-stranded DNA viral elements.

## Introduction

The recent boom of genome sequencing programs has revealed the abundance of DNA fragments of viral origin within eukaryotic genomes. These so-called Endogenous Viral Elements (EVEs) stem from endogenization events that not only involve retroviruses as donors (as could be expected from their natural lifecycle) but also viruses that do not typically integrate into their host chromosomes (*Katzourakis and Gifford, 2010*; *Feschotte and Gilbert, 2012*; *Aswad et al., 2021*). In insects, where retroviruses have yet to be found, endogenization events have involved various non-retroviral viruses: three families of large double-stranded (ds) DNA viruses, at least 22 families of RNA viruses, and three families of single-stranded (ss) DNA viruses (*Gilbert and Belliardo, 2022*). Degeneracy and loss are likely the fate of most EVEs, since they do not a priori benefit their hosts. Still, several studies have reported that EVEs can be retained by selection, thus becoming *domesticated* (*Koonin and Krupovic, 2018*). The functions involved include defensive properties against related viruses in mosquitoes (*Yan et al., 2009*; *Suzuki et al., 2020*), against macroparasites in some Lepidoptera (*Gasmi et al., 2021*), or modifications in the expression of genes involved in dispersal in aphids (*Gasmi et al., 2021*; *Parker and Brisson, 2019*). Beyond insects, the membrane fusion capacity of viruses, allows their entry into host cells, have been repeatedly co-opted in three metazoan clades: mammals, viviparous lizards, and parasitoid wasps. In placental mammals and viviparous Scincidae lizards, domestication of the *syncytin* protein from retroviruses has allowed the emergence of the placenta, through the development of the syncytium (composed of fused cells) involved in metabolic exchanges between the mother and the fetus (*Lavialle et al., 2013*; *Cornelis et al., 2017*). A similar fusogenic property was repeatedly co-opted by parasitoids belonging to the Hymenoptera order through the endogenization and domestication of complex viral machineries deriving from large dsDNA viruses (*Drezen et al., 2017*; *Gilbert and Belliardo, 2022*). The numerous retained viral genes allow parasitoid wasps to produce virus-like structures (VLS) within their reproductive apparatus. These are injected into the wasp's host, together with their eggs, and protect the wasp progeny against the host immune response. This protection is achieved thanks to the ability of VLS to deliver virulence factors in the form of genes (in which case VLS are called polydnavirus - PDV) or proteins (in which case VLS are called Virus-like particles - VLPs) to host immune cells (reviewed in *Gauthier et al., 2018*; *Drezen et al., 2022*). So far, five independent cases of such viral domestication have been detected in parasitoid wasps, four of them falling within the Ichneumonoidea superfamily (*Bézier et al., 2009*; *Volkoff et al., 2010*; *Pichon et al., 2015*; *Burke, 2019*) and one in the Cynipoidea superfamily (*Di Giovanni et al., 2020*). The four cases where the donor virus family has been unequivocally identified point towards dsDNA viruses. More specifically, the domesticated EVEs (hereafter, dEVEs) derive from the *Nudiviridae* family in three cases (*Bézier et al., 2009*; *Pichon et al., 2015*; *Burke, 2019*) while the fourth involves a putatively new viral family denoted 'LbFV-like' (*Di Giovanni et al., 2020*). Notably, all these domestication events took place in endoparasitoids, that is, in species that deposit their eggs inside the hosts, as opposed to ectoparasitoids that lay on their surface.

Beyond these well-characterized events of viral domestication in Hymenoptera, additional cases of endogenization have been uncovered, in studies that enlarged the taxonomic focus of either the hosts (*Ter Horst et al., 2019*; *Cheng et al., 2020*; *Kondo et al., 2019*) or the viruses that were considered (*Flynn and Moreau, 2019*; *Ter Horst et al., 2019*; *Irwin et al., 2022*; *Li et al., 2022*). Here, we complement this earlier work by expanding the range of both the hosts and viruses under study, and by further analyzing which endogenization cases have been followed by a domestication event.

To this end, we developed a bioinformatic pipeline to detect endogenization events involving any kind of viruses (DNA/RNA, single-stranded, double-stranded), at the scale of the whole Hymenoptera order. This analysis first allowed us to test whether the propensity of viruses to enter Hymenoptera genomes, and to be domesticated, depend on their genomic structure (in line with the pattern observed so far, where only dsDNA viruses have been involved in domestication events as described above). We then tested whether the lifestyle of the species (free-living, endoparasitoid, ectoparasitoid) correlates with their propensity to integrate and domesticate viruses. Our working hypothesis was that the endoparasitoid lifestyle may be associated with a higher rate of viral endogenization and/or a higher rate of domestication events, for two non-exclusive reasons related either to the exposure to new viruses and the adaptive value of the endogenized elements.

First, a higher endogenization rate may simply stem from a higher exposure to viruses. Such an effect could be at play in endoparasitoids due to the intimate interaction between the parasitoid egg

or larva and the host. In other words, the endoparasitic way of life may facilitate the acquisition of new viruses deriving from the hosts. Notably, this lifestyle may also facilitate the maintenance and spread of newly acquired viruses within wasp populations. Indeed, endoparasitoid wasps often inject not only eggs but also venomic compounds (typically produced in the venom gland or in calyx cells) where viruses can be present and may thus be vertically transmitted (*Martinez et al., 2016*). In addition, the confinement of the several developing wasps within a single host may facilitate viral horizontal transmission and its subsequent spread in wasps populations (e.g. *Varaldi et al., 2003*).

Second, a higher rate of domestication in endoparasitoids may be the consequence of a particular selective regime. This is expected since, these insects are facing the very special challenge of resisting the host immune system, contrary to other lifestyles. This selective pressure may promote the co-option of viral functions such as the above-mentioned membrane fusion activity, that provide a very effective mean to deliver virulence factors.

Our analysis reveals numerous new instances of endogenization events, some of which are also characterized by signatures of molecular domestication. We found a clear enrichment in endogenization events deriving from dsDNA viruses as compared to those with other genomic structures. While the data did not reveal a significant effect of Hymenoptera lifestyles on the acquisition of dsRNA, ssRNA, or ssDNA viruses, it supports the hypothesis that genes from dsDNA viruses are more often endogenized and domesticated in endoparasitoids than in free-living and ectoparasitoid species.

## Results

We screened for EVEs 124 Hymenoptera genome assemblies, including 24 ectoparasitoids, 37 endoparasitoids, and 63 free-living species (the list can be found in *Supplementary file 2*). EVEs were identified using a sequence-homology approach based on a comprehensive viral protein database. Different confidence levels (ranging from A to D) were associated with the various EVEs inferred, where the A score indicates a maximal confidence level for endogenization. This confidence index is based upon sequencing depth combined with the presence of eukaryotic genes and/or transposable elements in the genomic environment of the candidate loci (as detailed in the Material and methods section). By default, the four categories are included in the analysis, but unless otherwise stated statistical tests based on the A category only led to the same conclusions (see *Figure 1—figure supplements 2–7* for more details). Since several EVEs may enter into the genome during a single endogenization event, we grouped into the same event EVEs that were localized in the same scaffold (only for viruses having similar genomic structure), and/or that derived from the same putative viral family. Our analysis further included an inference of the phylogenetic relationships among homologous EVEs, that was used to map endogenization events on the Hymenoptera species tree. Finally, inferences of domestication events relied upon signatures of purifying selection in the integrated genes (based on *dN/dS* estimates) and/or on expression data. An important objective of our analysis is to detect and enumerate not only EVEs but also endogenization *events* that can explain the presence of these EVEs. Indeed, an EVE denotes a single gene of viral origin in a single species. Several neighboring EVEs in a genome most likely result from the endogenization of a single viral genome, and homologous EVEs shared by several closely related species may further stem from a single ancestral endogenization event. This distinction is critical when it comes to examining the effect of various factors on the probability of integrating EVEs, which implies counting events rather than EVEs. As an example, let's consider the single endogenization event involving 13 EVEs that occurred in the common ancestor of *Leptopilina* species (*Di Giovanni et al., 2020*). In this wasp genus, based on previous findings, we expect the 39 EVEs to be grouped into a single endogenization event. Our pipeline appropriately detected 36 EVEs (out of 39) and correctly aggregated them into a single endogenization event mapped on the branch leading to the *Leptopilina* genus. Thus, in *Figure 1*, we can observe a pie chart at the node corresponding to the common ancestor of the three *Leptopilina* species (*L.boulardi*, *L.clavipes,* and *L.heterotoma*). Most of this pie chart is blue, which corresponds to the putative donor viral family, i.e., LbFV-like, and is surrounded by a black border, indicating that the genes involved are inferred as domesticated. The number of EVEs and dEVEs (n=12/13) for each of the 3 species is then plotted along the horizontal bar plots with the same color code (see *Figure 1* and *Figure 1—figure supplement 3* for more canonical examples).

In total, the pipeline correctly detected 88.4% (152/172) of the EVEs involved in our four 'positive controls,' previously described as mediating the protection of young wasps against their host

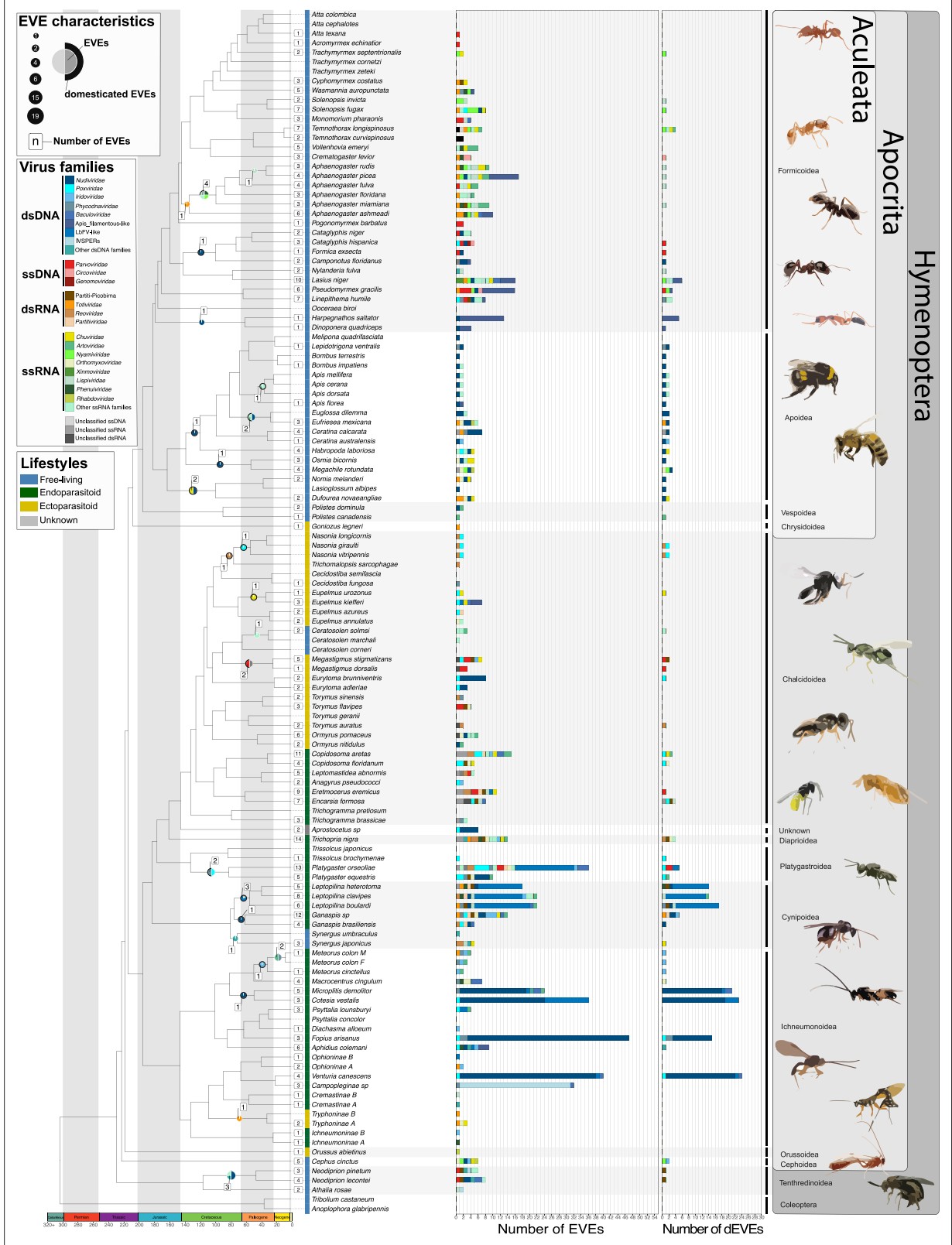

**Figure 1.** Endogenous Viral Elements (EVEs ) and their domestication status in Hymenoptera. Lifestyles are displayed next to species names (blue: free-living, green: endoparasitoid, yellow: ectoparasitoid, gray: unknown). The number of EVEs and domesticated EVEs (dEVEs) found in each species are represented respectively by the first and second facets of the horizontal histograms. Colors along these histograms indicate the potential donor viral families (where blue tones correspond to viral double-stranded DNA (dsDNA) viruses, red tones to single-stranded DNA (ssDNA) viruses, orange/

*Figure 1 continued on next page*

*Figure 1 continued*

brown tones to dsRNA viruses, and green tones to ssRNA genomes). EVEs shared by multiple species and classified within the same event are represented by circles whose size is proportional to their number; those that are considered as dEVEs are surrounded by a black border. Numbers in the white boxes correspond to the number of endogenization events inferred. As an example, *Megastigmus dorsalis* and *Megastigmus stigmatizans* are ectoparasitoids (yellow) sharing a common endogenization event (within the Cluster21304, see *Figure 1—figure supplement 1*) that likely originated from an unclassified dsRNA virus (gray color in circle), and shows no sign of domestication (no black border around the gray part of the circle). The figure was inspired by the work of *Peters et al., 2017*. Details on the phylogenetic inference and time calibration can be found in the Material an methods section; bootstrap information can be found in *Figure 1—figure supplement 7*; details on lifestyle assignation can be found in *Supplementary file 2*. All Cluster sequence alignments from loci scored from A to X can be found within the *Figure 1—source data 1*.

The online version of this article includes the following source data and figure supplement(s) for figure 1:

**Source data 1.** File including all aligned cluster sequences scored from A to X.

**Figure supplement 1.** Example of endogenization events.

**Figure supplement 2.** Simplified summary of the bioinformatics pipeline for the detection and validation of candidates for endogenization and domestication.

**Figure supplement 3.** Canonical examples of endogenization events inferred by our pipeline.

**Figure supplement 4.** IVSPER genes identified in the Campopleginae genome.

**Figure supplement 5.** Cladogram of the Ophioniformes group, illustrating the two independent endogenization events of two unknown viruses in Banchinae and Campopleginae lineages.

**Figure supplement 6.** Ultra conserved element (UCE) trees built to assign to species the unknown Chalcidoidea sequenced with the pool of *P. orseoliae*.

**Figure supplement 7.** Source of the datasets and availability of the reads.

**Figure supplement 8.** Representation of the score distribution among different virus genome types.

**Figure supplement 9.** Heatmap representing the viral genes known to be domesticated by Hymenoptera.

immune system. Among them, 71.82% were inferred as being domesticated. Out of the 152 positive controls EVEs, 147 were grouped into four independent endogenization events, as was expected. The remaining five genes had peculiar histories that led our pipeline to infer two additional spurious events (*Supplementary file 3*). All detailed results regarding EVEs and dEVEs can be found in the *Supplementary file 4*.

## Endogenizations involve all viral genomic structures

A total of 1261 EVEs have been inferred in the whole dataset (*Supplementary file 1*, *Figure 1*). These come from 367 endogenization events, the majority of which involved ssRNA and dsDNA viruses (41% and 35%, respectively) (*Supplementary file 1*). Among the 124 species under study, 113 underwent at least one endogenization event, with a maximum of 14 events and a median of 3 (*Figure 1*). In total, 91% of the events (331) occurred on tip branches, and the remaining 9% are shared by at least two closely related species (*Supplementary file 1*, *Figure 1*). To assess the validity of the procedure used to aggregate multiple EVEs into a single shared ancestral endogenization, we assessed whether EVEs inferred as homologous shared a common genomic environment. We thus tested for the presence of homologous loci in descendant species around the shared ancestral EVEs (using blastn searches between the corresponding scaffolds, see details in Materials and methods). Among the 36 endogenization events that involved at least two species, 31 were found to carry more homologous loci around the insertion sites than expected by chance (see details in Materials and methods). Notably, the majority of endogenization events involved a single EVE (a single gene) and only 12 (all from dsDNA viruses), involved the concomitant integration of more than four viral genes (*Figure 2C*) .

A total of 40 different viral clades (usually families) were inferred as putative donors. Most of them (34) are known to infect insects (*Figure 2B*) and these account for the majority of the 331 endogenization events. However, we found 36 EVEs (24 endogenization events), including 20 high-confidence ones (A-ranked), that derived from six viral families not previously reported to infect insects (*Phycodnaviridae*, *Herpesviridae*, *Caulimoviridae*, *Asfaviridae*, *Bornaviridae,* and *Mypoviridae*). However, in those cases, the true viral donors may belong to unknown clades that do infect insects. Indeed, although the homology with viral proteins was convincing (median e-value was 9.4095e-12 [min = 9.212e-129, max = 3.305e-08]), the average percentage identity was relatively low (38% [min = 23.2%,

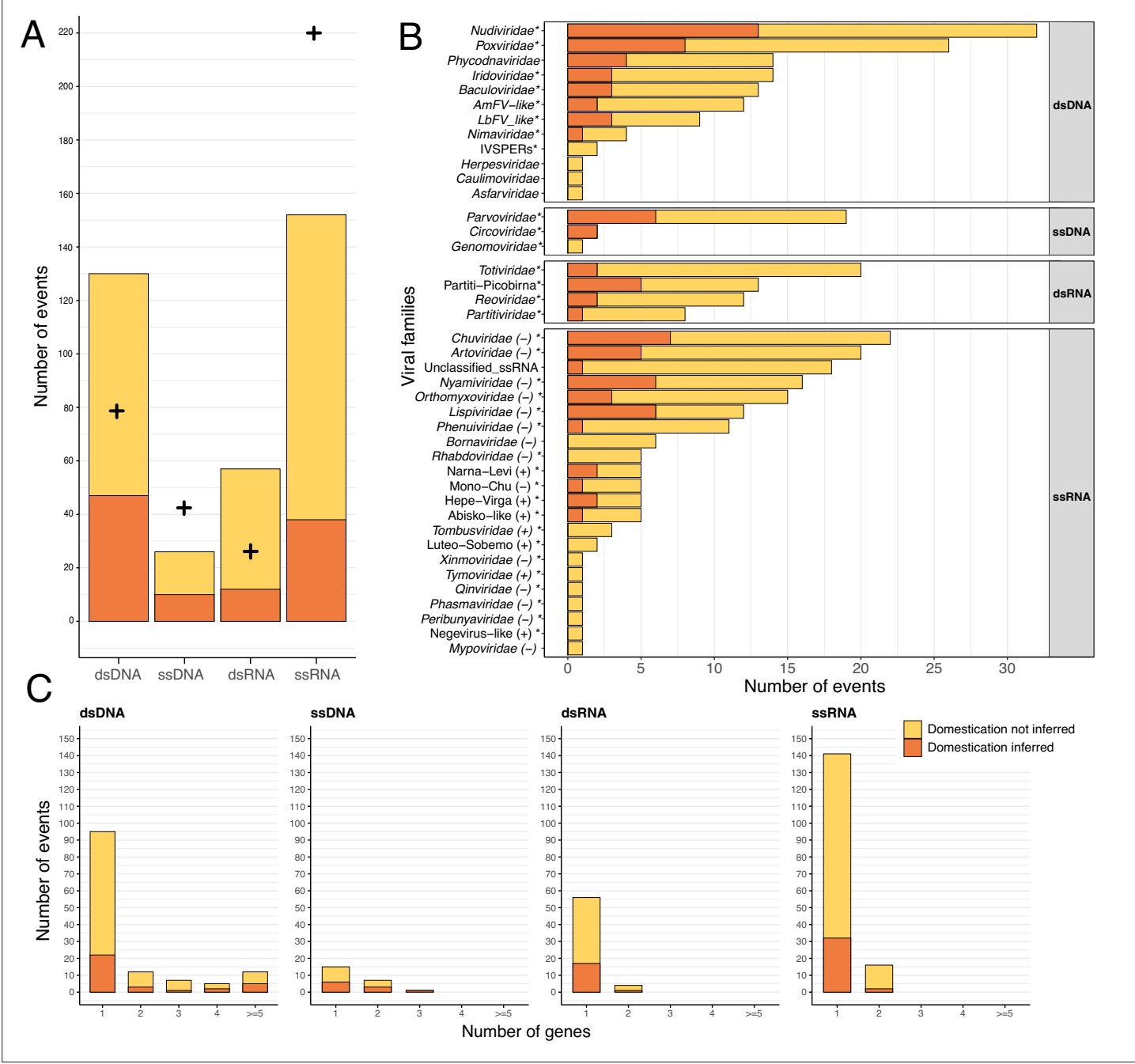

**Figure 2.** Endogenization involves all types of viral genomic structures. In all three panels, events inferred as corresponding to domestications are displayed in orange, while events not inferred as domestications are displayed in yellow. (**A**) Distribution of the number of events inferred, according to the four categories of viral genomic structures. The crosses refer to the expected number of endogenization events for each category based on its estimated relative abundance in insects (see details in Materials and methods and virus-infecting data in *Supplementary file 5*). The data used in this figure can be found in *Supplementary file 6* in the 'Figure_data' sheet. (**B**) Distribution of the various viral families involved in endogenization events. The polarity of single-stranded RNA (ssRNA) viruses is displayed next to the family name. Events involving multiple putative families (i.e. where several viral families are present in the same scaffold) have been excluded from the count. The star next to the family name indicates that the viral family is known to infect insects. (**C**) Distribution of the number of endogenous viral elements (EVEs) per event across viral categories.

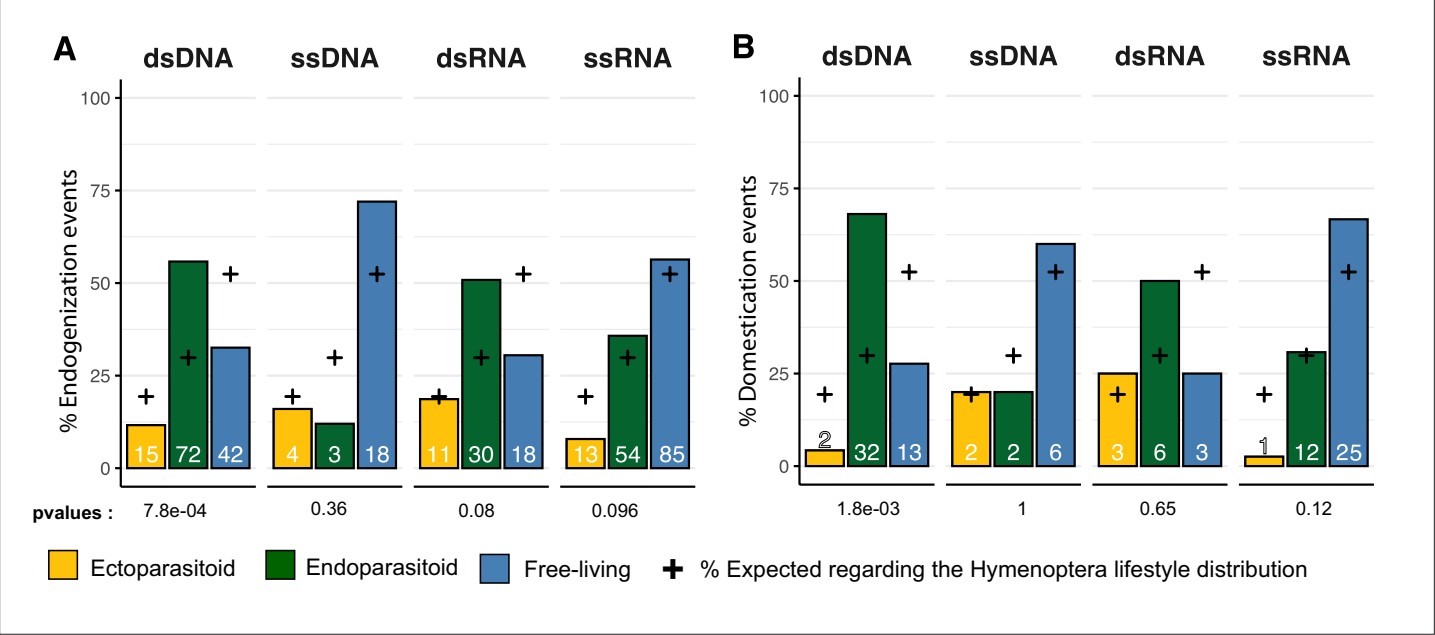

**Figure 3.** Endogenization and domestication of double-stranded DNA (dsDNA) viruses are most prevalent in endoparasitoid species. (**A**) Distribution of viral endogenization events (Event) and **B** of domestication events (dEVEs) across Hymenoptera lifestyles. Crosses indicate the expected proportion of events associated with the different lifestyles, based on the respective frequencies in our database (ectoparasitoid = 24/124, endoparasitoid = 37/124, free-living = 63/124). The p-values are the results of Fisher's tests comparing the observed and expected distributions. Numbers inside the bars indicate the absolute numbers of events inferred. The ancestral states of the nodes, in terms of lifestyle, were inferred in a Bayesian analysis (see details in Materials and methods). The data used in this figure can be found in *Supplementary file 6*.

max = 79.1%]), suggesting that these loci may originate from unknown viruses that are only distantly related to their closest relatives in public databases.

## dsDNA viruses are over-represented in endogenization events

Most of the endogenization events recorded involve ssRNA and dsDNA viruses. But do these proportions simply mirror the diversity and respective abundances of the different kinds of viruses encountered by insects? The analysis summarized in (*Figure 2A*) (see details in Materials and methods) indicates this is not the case. More specifically, it shows that dsDNA viruses are more frequently endogenized than expected on the basis of their representation in the database, while ssRNA viruses are under-represented ($\chi^2$=56.9 and 30.9, respectively, for endogenization events and domestication events, d.f.=3, both p-value <2.2e-7) (see data in *Supplementary file 6*). Notably, this result is not purely driven by the presence in our data set of the four positive controls (previously described cases of viral domestication, that all involve dsDNA viruses as donors, both p-value <2.2e-7). Finally, among endogenization events involving ssRNA viruses, we found an over-representation of negative-stranded ssRNA compared to their relative abundance in public databases (72.2% compared to 32.5% in the databases, $\chi^2$=147.29, d.f.=1, p-value <2.2e-16).

## Endogenizations of dsDNA viruses are more frequent in endoparasitoid species

Next, we sought to characterize the factors that could explain the patterns of endogenization events inferred (*Figure 1*). To this end, for each viral genomic structure, we assessed whether endogenization events were evenly distributed among the three wasp lifestyles, taking into account their respective frequencies in the dataset. No significant departure from the null hypothesis was detected for endogenization involving ssDNA, dsRNA, or ssRNA viruses (Fisher exact test p-values BH corrected >0.05). On the contrary, we detected a highly significant enrichment of dsDNA viruses endogenization events in endoparasitoid species, and conversely a deficiency in free-living and ectoparasitoid species (corrected p-value = 7.8e-04, *Figure 3A*)(see data in *Supplementary file 6*).

To further test the apparent correlation between Hymenoptera lifestyle and the rate of endogenization events, we inferred ancestral lifestyles along the phylogeny using a Bayesian model (see details in Materials and methods). We then constructed a generalized linear model where the dependent variable is the number of endogenization events inferred on each branch, while branch length and lifestyle are the explanatory variables (see details in Materials and methods). Branch length was included as an additive effect to remove the expected effect of time on the number of endogenization events, thus allowing the decomposition of the remaining variance according to the lifestyle (free-living, ectoparasitoid or endoparasitoid).

We first tested whether the rate of endogenization events deriving from any virus (that is, regardless of their genomic structures) was structured by lifestyles, and found no significant effect (*Figure 4—figure supplement 1A* left side). We then split the dataset according to the genomic structure of the donor viruses. For RNA or ssDNA viruses, the analysis did not reveal evidence of a correlation between wasps' lifestyles and the rate of endogenization events (*Figure 4—figure supplement 1G,I and K*). On the contrary, in the case of dsDNA viruses, we found a highly significant effect of the wasp lifestyle: endogenization rates appear to be 2.47 times higher in endoparasitoids than in free-living species (89% CI [1.56–3.56], *Figure 4A*). The corresponding probability of direction (pd, an index representing the confidence in the direction of an effect) was equal to 99.9%. In contrast, ectoparasitoids did not differ from free-living species (*Figure 4A*). Accordingly, more than 98% of the MCMC iterations led to a higher coefficient value for endoparasitoids than for ectoparasitoids (so-called $P_{MCMC}$ in *Figure 4A*). This effect was consistently found using high-confidence scaffolds only (A-ranked scaffolds, *Figure 4—figure supplement 1C* middle side). We also carried out the same analysis without the 4 domestication cases previously mentioned in the literature (because including them in our data set could have skewed the results) and reached the same conclusion (*Figure 4—figure supplement 1E* middle and left sides). Overall, these results show that dsDNA viruses are more often endogenized in endoparasitoids than in free-living and ectoparasitoid species. All model summaries can be found in the *Supplementary file 6* in the 'GLM_lifestyle_EVEs_dEVEs' sheet.

## Domestications of dsDNA viruses are most prevalent in endoparasitoid species

We then investigated whether lifestyles may explain the abundance of domestication events. A simple Fisher's exact test approach revealed an enrichment in endoparasitoid species of domestication events involving dsDNA viruses (Benjamini-Hochberg adjusted p-value = 1.8e-03), whereas no deviation from the null hypothesis was detected for the other viral genomic structures (*Figure 3B*) (see data in *Supplementary file 6*).

We built upon the generalized linear models described above, in a Bayesian framework, to test whether lifestyle could also be a factor explaining the propensity of Hymenoptera to domesticate (and not simply endogenize) viral genes (see details in Materials and methods). We found that the domestication of dsDNA viruses are 3.68 times more abundant in endoparasitoids than in with free-living species (89% CI [1.72–6.17], pd = 99.9%, *Figure 4B*). This effect was also detected when only high-confidence candidates were considered (*Figure 4—figure supplement 1D* middle side), or if we removed the four known cases of domestication (*Figure 4—figure supplement 1F* left and middle side). In other viral categories, no convincing effect of the wasp lifestyle was detected (all pd <99%) (*Figure 4—figure supplement 1H and J*) except for a lower rate of domestication of ssRNA viruses in ectoparasitoids compared to other lifestyles (*Figure 4—figure supplement 1L*).

Two non-mutually exclusive hypotheses may explain the high frequency of dsDNA virus domestication in endoparasitoids. First, it may simply stem from the higher rate of endogenization outlined above: a higher rate of entry would overall translates into a higher rate of domestication. Second, it may result more specifically from differences in the rate at which viral elements are domesticated after being endogenized. To disentangle these hypotheses, we built a binomial logistic regression model in a Bayesian framework, focusing on events involving dsDNA viruses, and specifying the number of domesticated events *relative* to the total number of endogenization events inferred. By controlling for the endogenization input (the denominator), these binomial models make it possible to test whether the probability of domestication after endogenization of dsDNA viruses is correlated with the lifestyle.

Based on this analysis, the probability that an endogenization event will lead to a domestication event is not significantly different between endoparasitoids and free-living species (*Figure 4—figure*

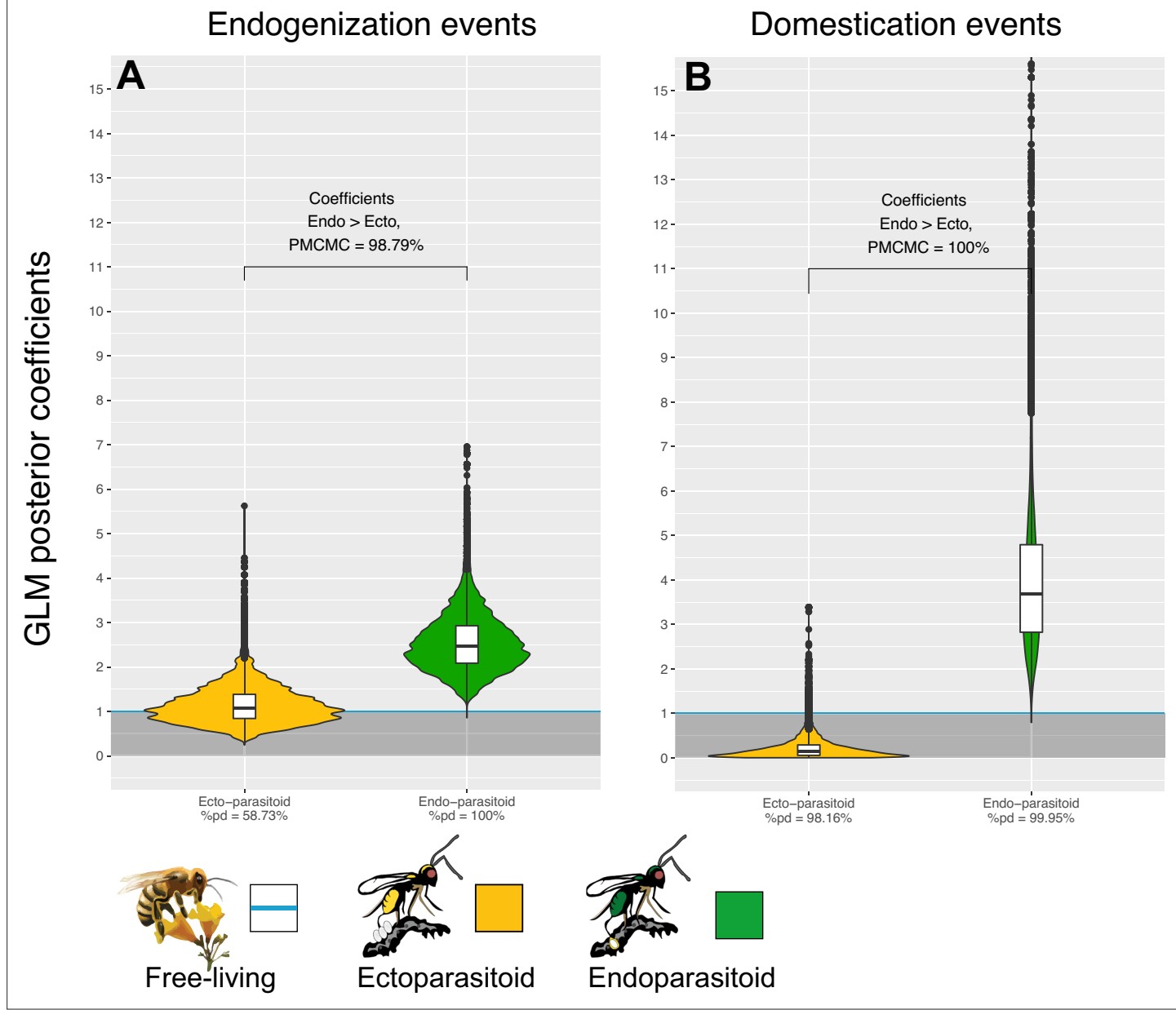

**Figure 4.** Endogenization and domestication of double-stranded DNA (dsDNA) viruses are more frequent in endoparasitoid species. Violin plots represent the posterior distribution of the coefficients obtained under the different GLM models (after exponential transformation to obtain a rate relative to free-living species). The coefficients are derived from 1000 independent GLM models, where 1000 probable scenarios of ancestral states at nodes were sampled randomly among the MCMCM iterations (see details in Materials and methods). Branches from nodes older than 160 million years were removed from the dataset. The %pd is the probability of direction and indicates the proportion of the posterior distribution where the coefficients have the same sign as the median coefficient. $P_{MCMCM}$ indicates the proportion of MCMC iterations where the coefficient obtained for endoparasitoid species is higher than for ectoparasitoid species. All statistical summaries of the Bayesian GLM models can be found in **Supplementary file 6**.

The online version of this article includes the following figure supplement(s) for figure 4:

**Figure supplement 1.** Violin plots of the posterior distribution of GLM coefficients in relation to Hymenoptera lifestyle.

**Figure supplement 2.** Violin plots of the posterior distribution of dEvents GLM coefficients in relation to wasp lifestyle (corrected for Events rates).

**Figure supplement 3.** Violin plots of the posterior distribution of GLM coefficients in relation to Hymenoptera lifestyle.

*supplement 2A*, pd = 89.18%). However, the probability of domestication was found to be significantly higher in endoparasitoids than in ectoparasitoids (*Figure 4—figure supplement 2A*, $P_{MCMC}$=99.81%). The same trend was observed if we focused on high-confidence scaffolds and/or if we removed the four known controls from the dataset (*Figure 4—figure supplement 2B,C and D*, pd <86%).

Together, these findings show that the endoparasitoid lifestyle is associated with an increased rate of dsDNA viruses endogenization. Endoparasitoids are also characterized by an elevated frequency of domestication events that does not appears to be explained by an elevated rate of post-endogenization domestication.

## Conclusions hold when eusociality is taken into account

Our analysis relied on a simplification of Hymenoptera lifestyles (free-living, endoparasitoid, ecto-parasitoid). We may thus wonder whether adding complexity to this view would change our main conclusion. In particular, a significant proportion of our free-living species are in fact eusocial (ants, some bees). We thus tested the robustness of our conclusion by including an 'eusociality' category in our analysis (see new categories in *Supplementary file 2*).

First, the new models did not reveal any significant effect of the lifestyle in the number of Events or domestication Events involving ssRNA and ssDNA viruses (*Figure 4—figure supplement 3G,H and K,L*). Second, for dsRNA viruses, eusocial insects had a reduced number of dEvents compared to free-living, ecto, or endoparasitoid species(pd >99%) (*Figure 4—figure supplement 3I,J*). Most importantly, this analysis revealed again a greater number of endogenization events or domestication events involving dsDNA viruses in endoparasitoid compared to free-living species (pd >99%, *Figure 4—figure supplement 3C,D*). The conclusions remained unchanged when controls were removed from the dataset (*Figure 4—figure supplement 3E and F*). When we only considered loci annotated with an A score (highly confident EVEs), we found the exact same pattern, although the effect was only marginally significant for the number of events (pd = 98.6%) (*Figure 4—figure supplement 3C and D*). In addition, more than 98% of the MCMC iterations led to a higher coefficient value for endoparasitoids compared to ectoparasitoid or eusocial species in that combination.

When combining both filters (only A score and without the controls), the same tendency was observed, although the effects were only marginally significant (pd min = 96%) (*Figure 4—figure supplement 3E and F*). In addition, more than 99% of the MCMC iterations led to a higher coefficient value for endoparasitoids compared to ectoparasitoid or eusocial species in that combination. In conclusion, when a eusocial category is added, the same conclusion is reached: dsDNA viruses are more often endogenized and domesticated in endoparasitoids than in free-living, ectoparasitoid, and eusocial species.

## New remarkable cases of endogenization and domestication

Here, we describe in more detail specific cases identified by our pipeline. We found a massive entry of genes from dsDNA viruses in an undescribed species belonging to the Campopleginae subfamily ('Campopleginae sp' in *Figure 1*). In Ophioniformes (a clade that includes Campopleginae), two lineages that have previously been shown to host domesticated viruses (the Campopleginae species *Hyposoter didymator* (*Volkoff et al., 2010*), and the Banchinae species *Glypta fumiferanae Béliveau et al., 2015*). It has been advocated that these so-called ichnoviruses found in *Hyposoter didymator* and *Glypta fumiferanae* may derive from the same endogenization event (*Béliveau et al., 2015*). In our unknown Campopleginae species, we identified homologs of 35 out of the 40 ichnovirus genes present in the genome of *H. didymator* (so-called IVSPER genes, *Volkoff et al., 2010*). Those genes show conserved synteny in the two species (*Figure 1—figure supplement 4*), strongly suggesting that they derive from the same endogenization event. However, our analysis did not identify viral homologs in the two Ophioninae and Cremastinae subfamilies, that are internal to the clade including Campopleginae and Banchinae wasps. Together with previous studies that does not reveal the presence of IVSPER in other internal clades, this result argues against the view of a single event at the root of Ophioniformes, and supports the alternative view (*Burke et al., 2021*) that the so-called IVSPER genes in the Campopleginae and Banchinae subfamilies stem from independent events, despite their striking structural similarities (see *Figure 1—figure supplement 5* for illustration). We found no trace of the previously suggested remnants of ichnoviruses in the related species *Venturia canescens* (*Pichon et al., 2015*), whereas the presence of nudiviral genes in this species was confirmed.

Additionally, we found five new cases of endogenization involving multiple EVEs from dsDNA viruses belonging to *Nudiviridae*, LbFV-like, and AmFV-like families.

Two of them involve parasitoid species, i.e., *Platygaster orseoliae* and an *Aprostocetus* species. For *Aprostocetus*, we detected six EVEs related to nudiviruses branching between the Chalcidoidea and the Diaprioidea superfamilies (*Figure 5—figure supplement 1D*). Among these EVEs we found four with an annotation: *lef-4*, *Ac68/pif-6*, GrBNV_gp19/60/61-like proteins, and a rep-like protein. In the absence of closely related sequences or RNA seq data, we cannot investigate if these elements have been domesticated.

The *P. orseoliae* case involves the recently characterized putative family of filamentous viruses (*Lepetit et al., 2016*). The free-living LbFV virus is the only representative of this putative family and has been identified as a source of adaptive genes in *Leptopilina* wasps that parasitize *Drosophila* flies, with 13 virally-derived genes involved in the production of VLPs protecting the wasp's eggs from encapsulation (*Di Giovanni et al., 2020*). In *P. orseoliae*, 15 genes homologous to LbFV were detected (out of 108 ORFs in the LbFV genome; median E-value=9.39e-21 [min = 2.617e-76, max = 4.225e-08]) (*Figure 5—figure supplement 2A*). Among these 15 genes, five were also endogenized in *Leptopilina* species (named LbFV_ORF58:DNApol, LbFV_ORF78, LbFV_ORF60:LCAT, LbFV_ORF107, and LbFV_ORF85) (*Di Giovanni et al., 2020*). Assuming the ancestral donor virus contained the same 108 genes as LbFV, the number of shared genes in these two independent domestication events is higher than expected by chance (one-sided binomial test: x=5, n=15, p=13/108, p=0.02682), suggesting that similar functions could have been retained in both lineages. Notably, we also found within the *P. orseoliae* assembly 12 scaffolds that were annotated as free-living viruses (F-X scaffolds). They had a different sequencing depth compared to BUSCO containing scaffolds and encoded 136 complete ORFs for which 21 presented homology with LbFV ORFs (min bit score = 50, min ORF size = 150 pb, max overlaps = 23 pb). ORF density was 82.7% which is in the range of expected values for related free-living viruses. In order to identify additional scaffolds possibly belonging to this free-living virus, we searched for homology between the 136 putative viral proteins, and the scaffolds obtained from the assembly of *P. orseoliae*. These results allowed us to identify two additional scaffolds (scaffold_117128 and scaffold_18896). Because the total size of the 14 putative 'free-living' scaffolds were within the expected range for a dsDNA virus genome (136,801 bp) and because the average coverage was much higher than BUSCO-containing scaffolds (mean cov = 95.6 X [sd = 5.05 X] compared to 33 X in BUSCOs) and homogeneous (*Figure 5—figure supplement 1A*), we believe that this set of scaffolds corresponds to the whole genome of a new virus, related to LbFV, which we propose to call Platygaster orseoliae filamentous virus (PoFV) (see *Figure 5*). This virus is the closest relative to the EVEs found in *P. orseoliae*. Using this putative whole genome viral sequence to search for homologous genes in the *P. orseoliae* genome, we were able to detect a total of 139 convincing EVEs (deriving from 89 of the 136 ORFS found in PoFV). A large proportion of the EVEs (22.7%) presented premature stop codons within the sequences, further suggesting that these virally-derived genes are indeed endogenized since abundant pseudogenization is not expected in free-living virus genomes (*Figure 5—figure supplement 2A*). 44 of these 89 PoFV-derived EVEs presented signs of domestication, as the *dN/dS* (inferred using paralogs) were significantly lower than 1 (see *Ac81* and *Integrase* gene phylogenies in *Figure 5—figure supplement 3*). Among the 81/139 apparently intact EVEs (with ORF length >= 50% of the PoFV ORF), some are likely implicated in DNA replication (integrase), in transcription (lef-8, lef-9, lef-5, lef-4), in packaging and envelopment (ac81, 38 k) and in infectivity (pif-1, pif-2, pif-3). Among the 139 PoFV-related EVEs found in *P. orseoliae*, 104 corresponded to putative paralogs whereas none of these 104 ORFs were present in multiple copies within the PoFV genome, suggesting that a major post-endogenization duplication event occurred or that multiple endogenization events did occur. Although functional studies are clearly needed to confirm that these virus-derived genes are involved in the production of VLPs as in *Leptopilina* (*Di Giovanni et al., 2020*), we see Platygaster orseoliae endogenous viral elements (PoEFVs) as good candidates for viral domestication, which could possibly be involved in counteracting the immune system of its dipteran host (from the Cecidomyiidae family *Buhl and Hidayat, 2016*). To our knowledge, this is the first report of a massive viral endogenization and putative domestication within the *Platygastroidea* superfamily.

The other three cases involved ant species: *Harpegnathos saltator* (EsEFV) (12EVEs/6dEVEs), *Pseudomyrmex gracilis* (PgEFV) (9EVEs/1dEVE) (*Figure 5—figure supplement 1C* and *Figure 5—figure supplement 4*), *Aphaenogaster picea* (ApEFV) (7EVEs) (*Figure 5—figure supplement 4*). These

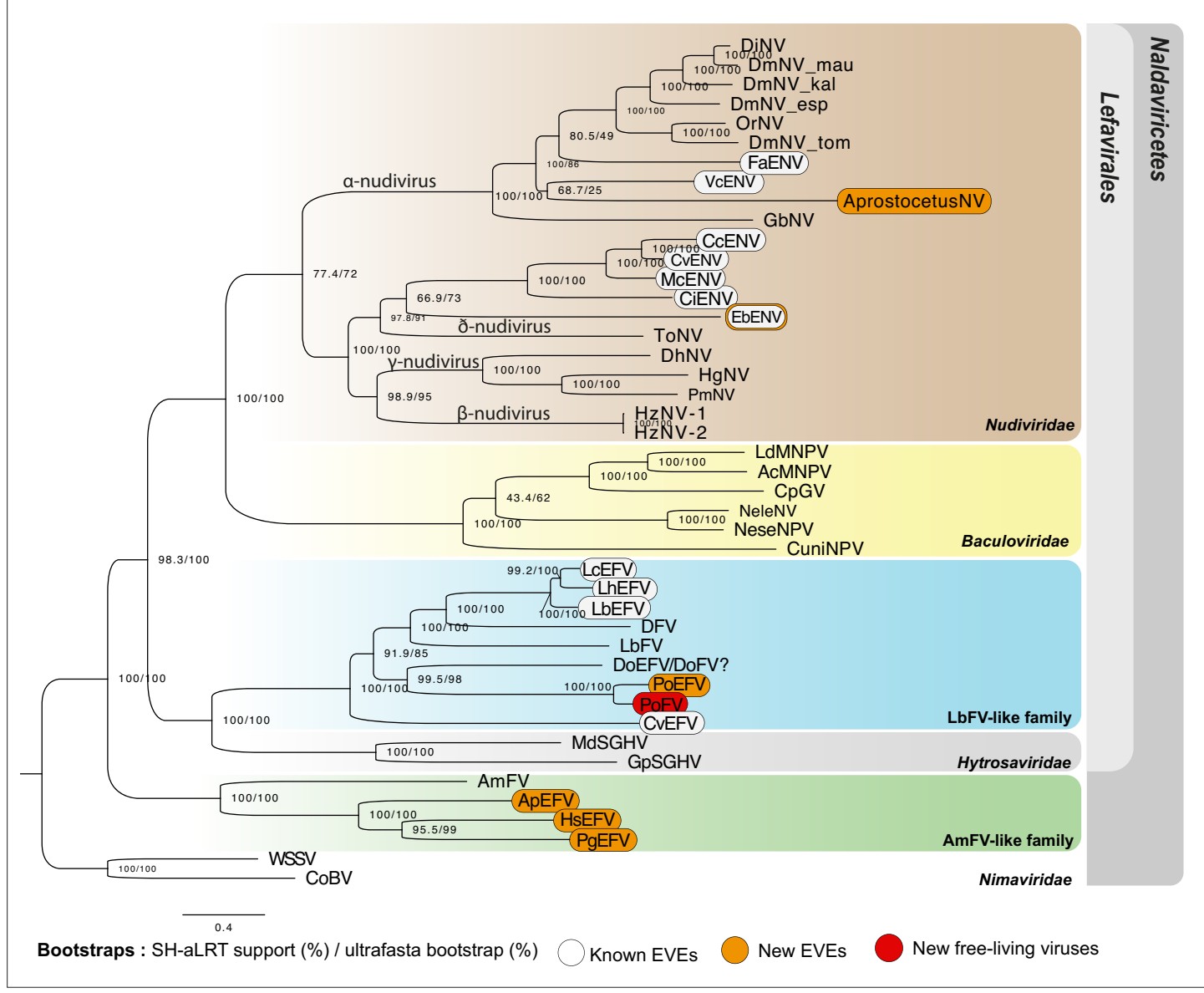

**Figure 5.** Phylogenetic relationships among endogenized and 'free-living' double-stranded DNA (dsDNA) viruses. Specifically, this figure shows the relationships between *Naldaviricetes* double-stranded DNA viruses and endogenous viral elements (EVEs) from hymenopteran species, where at least three endogenization events were found. This tree was computed using maximum likelihood in Iqtree (v2) from a 38,293-long protein alignment based on the concatenation of 142 viral genes. Confidence scores (aLRT%/ultra-bootstrap support%) are shown at each node. The scale bar indicates the average number of amino acid substitutions per site. Previously known EVEs are in white, those from the present study in orange, and leaves inferred as free-living viruses are in red. All the best partitioned models as well as the number of genes used for each taxa can be found in *Supplementary file 7*. All free-living dsDNA viruses used in this phylogeny were obtained from published complete viral genomes. More details on the phylogenetic inference can be found in the methods. The '?' for DoEFV/DoFV refers to the uncertainty regarding the endogenous/exogenous status of this sequence. All Cluster sequence alignments can be found in the *Figure 5—source data 1*.

The online version of this article includes the following source data and figure supplement(s) for figure 5:

**Source data 1.** File including all aligned cluster sequences included in the *Naldaviricetes* phylogenetic analysis.

**Figure supplement 1.** G+C% Coverage distribution of scaffold containing multiple endogenous viral elements (EVEs) Events.

**Figure supplement 2.** Genomic environment for the endogenous viral elements (EVEs) detected in *Platygaster orseoliae*.

**Figure supplement 3.** Phylogenies of LbFV-like proteins under purifying selection in *Platygaster orseoliae* genome.

**Figure supplement 4.** Candidate endogenous viral elements (EVEs) in two ant species.

endogenized elements are related to a poorly characterized family of filamentous viruses denoted AmFV (*Hartmann et al., 2015*; *Yang et al., 2022*). In *H. saltator*, nine genes deriving from an AmFV-like virus were detected (including three genes that have been previously identified by *Flynn and Moreau, 2019*). Intriguingly, all these genes presented numerous paralogs within the genomes (135 in total) (*Figure 5—figure supplement 1B*), with 22 copies for AmFV_0062 (*pif-1*), 18 for AmFV_0102 (*pif-2*), 51 for AmFV_0090 (*pif-3*), 24 for AmFV_0044 (*integrase*), 13 for AmFV_0079 (*p74*), five for AmFV_0047 (*RNA polymerase Rpb1, domain 2*), 19 for AmFV_0126 (Unknown), 23 for AmFV_0168 (Unknown), and seven for AmFV_0154 (Unknown). Most paralogs were found in scaffolds exceeding the expected size for any virus sequence (min = 23,726 bp, mean = 326,262 bp, max = 2,693,376 bp). In addition, all scaffolds do include transposable elements and eukaryotic genes making them undoubtedly endogenized. Accordingly, our pipeline attributed the highest confidence index A for 104 of them (out of 135). The *P. gracilis* genome revealed nine EVEs, including homologs of *pif-1*, *pif-3*, *RNA polymerase*, *ac81*, *integrase,* and *odv-e56* (*Figure 5—figure supplement 4*). Notably, one of the 9 EVEs (AmFV_059, of unknown function) shows both a *dN/dS* <1 (mean = 0.1747, p-value 5.877e-02), and a very high TPM value (362,836 TPM from whole body tissues). Finally, in *A. picea*, seven EVEs were detected, including homologs of *pif-1*, *pif-3*, *integrase*, *odv-e56,* and *p74* (*Figure 5—figure supplement 4*). No raw reads data were available for this species, precluding coverage-based inferences. Since there were neither orthologs or paralogs for these genes to compute *dN/dS* analyses, nor transcriptomic data, it was not possible to infer their domestication status. At this stage, it is thus not possible to conclude as to the functions of these genes in *H. saltator*, *P. gracilis,* and *A. picea*, but this surely deserves further attention.

## Discussion

All kinds of viruses can integrate arthropod genomes, although the mechanisms underlying these phenomena remain unclear (*Katzourakis and Gifford, 2010*; *Gilbert and Belliardo, 2022*). Prior to the present analysis, 28 viral families had been described as involved in endogenization in arthropods (*Gilbert and Belliardo, 2022*). Our study of hymenopteran genomes which relied on sequence homology search, further revealed the ubiquity of this phenomenon, with 1261 EVEs found, belong to at least 40 viral families (or family-like clades). Because the identification of EVEs necessitates the availability of related viruses in the database, we should see these numbers as an underestimation of the real number of EVEs. In addition, our pipeline necessitates the availability of either related species sharing the same EVEs (or at least the presence of paralogs within a single species) or the availability of RNAseq data to infer domestication. Because these last conditions were only met for 701 out 1261 EVEs, the results we obtained here regarding domestication should be seen as an underestimation of the prevalence of the phenomenon. Of the 1,261 EVEs found, the average identity with the closest known viral proteins was 36.32% [min = 15.7%, max = 99.1%]. Although this large overall divergence suggests ancient events, it does not exclude the possibility that some of the integrations are recent, because free-living viral relatives of the true donors may be unknown or extinct (*Junglen and Drosten, 2013*).

In the following section, we will first discuss a hypothesis for why double-stranded DNA viruses, in comparison with other viral genomic structures (ssDNA, dsRNA, ssRNA), are more often endogenized than expected. We will then discuss hypotheses that could explain why we found a higher rate of endogenization of dsDNA viruses in endoparasitoids compared to ectoparasitoids or free-living hymenopterans, which also translates into more frequent events of domestication.

### dsDNA viruses are more frequently involved in endogenization than expected by chance

Despite the observations that all viral genomic structures can be involved in endogenization, we clearly identified differences in their propensity to do so. Based on a comparison between the respective proportions of the various viral categories in the inferred endogenization events and in public databases, we found that dsDNA viruses are much more represented than expected, while ssRNA viruses are under-represented (*Figure 2A*). We acknowledge that current knowledge on the actual diversity of free-living viruses (as approximated through the NCBI taxonomy database) remains incomplete, but the strength of the effect reported here makes this conclusion rather robust to variations

in the null distribution. On the basis of current knowledge, RNA viruses, and in particular ssRNA viruses, appear to be much more diversified and prevalent than DNA viruses in insects. We note that viral-metagenomic studies often focus either on DNA or RNA viruses, and as such do not provide an accurate and unbiased picture of the extent of viral diversity. To gain insights on this topic, we may thus focus on model systems where long-lasting research efforts have likely produced a more reliable picture. The Honeybee *Apis mellifera* is probably the most studied of all hymenopteran species. In honeybees, the great majority of known viruses belong to the RNA world (*Chen and Siede, 2007*), with very few exceptions (*Yang et al., 2022*). Similarly, until 2015, only RNA viruses were known to infect the fruit fly *Drosophila melanogaster*, despite the extensive research conducted on this model system (*Webster et al., 2015*). A very limited set of DNA viruses has now been described from this species (*Wallace et al., 2021*) but clearly, RNA viruses dominate the *Drosophila* viral community, both in terms of diversity and prevalence. In support of this view, recent studies revealed the very elevated absolute diversity of RNA viruses. For instance, a survey of 600 insect transcriptomes recovered more than 1,213 RNA viruses belonging to 40 different families (*Wu et al., 2020*). Although, obviously, this study does not inform on the diversity of DNA viruses, it shows that the RNA virome of insects is both prevalent (e.g. in this study, 15% of all insects were infected by a single Mononegales-like virus) and extremely diversified (*Wu et al., 2020*). Actually, this view appears to hold at the larger scale of eukaryotes (*Koonin et al., 2015*). Taking into account this patent abundance of ssRNA viruses in insects, our study indicates they are by far less frequently endogenized than their dsDNA counterparts in hymenopterans. Notably, a similar trend was recently reported in a study including a diverse set of eukaryotes (*Irwin et al., 2022*).

Most of the major endogenization events characterized so far in hymenopterans involve dsDNA viruses from the *Nudiviridae* family (*Cheng et al., 2020*; *Burke, 2019*; *Pichon et al., 2015*; *Bézier et al., 2009*; *Cheng et al., 2014*; *Zhang et al., 2020*; *Gilbert and Belliardo, 2022*). Our study further confirms that this viral family represents a major source of exogenous and sometimes adaptive genes for Hymenoptera. Indeed, 28 new independent endogenization events involve in this family, among which 9 are shared by at least two related species (*Figure 2B*, *Figure 1*). The major contribution of nudiviruses to endogenization may be explained by their wide host range in arthropods (*Wang et al., 2007*). Their nuclear replication constitutes another plausible explanatory factor (*Velamoor et al., 2020*), since it may facilitate contact with host DNA. In addition, their tropism for gonads may favor the endogenization in germinal cells (*Burand, 2009*). In fact, nuclear replication is a feature shared by nearly all families of dsDNA viruses found in our analysis: *Baculoviridae*, *Iridoviridae*, *Phycodnaviridae*, *Nimaviridae Caulimoviridae*, *Herpesviridae*, *Asfaviridae* (although nuclear replication occurs only at early time post-infection) (*Schmid et al., 2014*; *Harrison et al., 2020*; *Alonso, 2023*; *Teycheney, 2023*; *Verbruggen et al., 2016*), Apis-filamentous-like (*Clark, 1978*) and LbFV-like families (*Varaldi et al., 2006*) (the *Poxviridae* viruses, that replicate in the cytoplasm, are thus the only exception). In contrast, most RNA viruses replicate in the cytoplasm. Nuclear replication may thus constitute a general explanation for the elevated propensity of DNA viruses to endogenization. Additionally, we may expect that a DNA molecule, rather than an RNA molecule, is more likely to integrate the insect genome, because the latter requires reverse transcription before possible endogenization.

The *Poxviridae* case indicates that cytoplasmic replication does not necessarily impede endogenization. These viruses do not require nuclear localization to propagate (*Moss, 2013*; *Schmid et al., 2014*) and were nevertheless found to be involved in many endogenization events (n=28). A similar pattern was observed in a recent study focusing on ant genomes (*Flynn and Moreau, 2019*). Within *Poxviridae*, entomopoxviruses were particularly involved in endogenization events (n=18) with four cases of EVEs shared between several closely related species (*Figure 2B*).

## Factors behind variation in endogenization and domestication rates

Several recent studies have uncovered abundant EVEs in insect genomes (*Flynn and Moreau, 2019*; *Ter Horst et al., 2019*; *Russo et al., 2019*), with huge variations in abundance between species. For instance, in their analysis based on 48 arthropod genomes, *Ter Horst et al., 2019* found that the number of EVEs ranged between 0 and 502. Although insect genome size and assembly quality may partly explain this variation (*Gilbert and Belliardo, 2022*), the underlying biological factors are generally unknown. In this study, we tested the hypothesis that the insect lifestyle may influence both the endogenization and domestication rates. We used a Bayesian approach to reconstruct ancestral

states throughout the phylogeny of Hymenoptera, thus accounting for uncertainty, and found that endoparasitoidism, in comparison with other lifestyles, tends to promote dsDNA viral endogenization. Notably, this conclusion was not the artefactual consequence of differences in genome assembly quality. In fact, the quality of genome assemblies was correlated with the lifestyle in our data set, but the genomes of endoparasitoid species were generally less well assembled than those of free-living species. If anything, this difference should reduce the power for detecting endogenization events in endoparasitoids, where our analysis detected an excess of such events. Our estimate of the effect sizes (with 2.47 times more endogenization events in endoparasitoids than in free-living species) should thus be seen as conservative.

Why do endoparasitoid wasps tend to undergo more endogenization than others? We initially had in mind two non-exclusive hypotheses that remain plausible explanations for the observed pattern. First, endoparasitoids may be more intensively exposed to viruses. However, if this effect is at play, we expect to have an 'endoparasitoid' effect for all viruses, whatever their genomic structure. For instance, we would expect such an effect to be detected for ssRNA viruses, which are involved in the greatest number of endogenization events (*Figure 2A*). This was not the case, since only dsDNA viruses were more frequently endogenized in endoparasitoids. Thus, we argue that this hypothesis is unlikely to explain the observed pattern. In addition, or alternatively, endoparasitoids may have a higher propensity to endogenize and retain viral genes. This second hypothesis posits that endoparasitoids are more frequently selected for retaining virally-derived genes than ectoparasitoid or free-living hymenopterans. In our analysis, domestication events are most frequently observed in endoparasitoids (over three times more frequently than in other hymenopterans). Obviously, this may be at least partly explained by the higher input discussed above (the higher endogenization rate). Yet, once this effect is controlled for, a trend towards a higher rate of domestication remains. More specifically, the likelihood of domestication following endogenization was significantly higher in endoparasitoids than in ectoparasitoids, but was not significantly higher than in free-living species. This latter lack of significant difference may be biologically explained if a single domestication event precludes the domestication of additional EVEs, while not affecting the rate of non-adaptive endogenization. This would 'dilute' the signal along branches involved in domestication. If this effect is at play, then it reduces considerably the power of our analysis to detect any difference in the rate of domestication between lifestyles. Indeed, in all known cases, only one domesticated virus has been documented, suggesting that further domestications are not beneficial once a viral machinery has been recruited by a wasp lineage.

Whether or not the rate of domestication *per se* is higher in endoparasitoids than in other hymenopterans, the selective advantages brought by these viral genes in endoparasitoids should be discussed. It has been demonstrated in a few model systems that EVEs may confer antiviral immunity against related 'free-living' viruses via the piRNA pathway (*Suzuki et al., 2020*; *Whitfield et al., 2017*). Yet, to our knowledge, such an effect has only been demonstrated against RNA viruses, so that it would not explain the excess of DNA viruses documented here. Furthermore, the sequence identities with known viral sequences, which is needed for this mechanism to work, is generally low in our dataset. Accordingly, previous work revealed that EVE-derived piRNAs studied in 48 arthropod species were also probably too divergent to induce an efficient antiviral response (*Ter Horst et al., 2019*). At that stage, the ability of EVEs to generate PIWI-interacting RNAs that play a functional role in antiviral immunity seems questionable. Further studies involving small RNA sequencing in hymenopterans would be required to shed light on this issue. Protection of the eggs and larvae against the host immune system is recognized as an important trait, where EVEs play a critical role. Because of their peculiar lifestyle, endoparasitoids are all targeted by the host immune system, a matter of life or death to which other hymenopterans are not exposed to. Several cases of endogenization and domestication in endoparasitoids, all involving dsDNA viruses, are thought to be related to this particular selective pressure (*Bézier et al., 2009*; *Volkoff et al., 2010*; *Pichon et al., 2015*; *Burke, 2019*; *Di Giovanni et al., 2020*). The parasitoids appear to have co-opted the viral fusogenic property to address their own proteins (VLPs) or DNA fragments (polydnaviruses) to host immune cells, thereby canceling the host cellular immune response. The above-hypothesized high exposure of endoparasitoids to viruses, together with this unique selective pressure, may act in concert to produce the pattern documented here: a strong input, that is, a diverse set of putative genetic novelties, combined with a strong selective pressure for retaining some of them. The observed excess of dsDNA viruses

may be an indication that these viruses display a better potential for providing adaptive material in this context. In the cases of polydnaviruses (found in some Braconidae and some Campopleginae), it appears that one way to efficiently deliver virulence factors to the host cell is by addressing DNA circles that ultimately integrate into the host immune cells and get expressed (*Chevignon et al., 2014*; *Chevignon et al., 2018*). The DNA which is packed into the mature particles typically encodes virulence proteins deriving from the wasp (*Espagne et al., 2004*). This means that, at least for these cases, the viral system should be able to pack DNA, which is most likely a feature that DNA viruses may provide. Such an argument does not hold in the VLP systems, where only proteins are packed in viral particles, and it is unclear why EVEs deriving from dsDNA viruses would be more able to fulfill such a function. Here other features of dsDNA viruses come into mind as possibly important factors: their large genome size, and their large capsids and envelopes (*Chaudhari et al., 2021*). These may predispose dsDNA viruses to be domesticated, since abundant quantities of venoms have to be transmitted in order to efficiently suppress the host immune response. In conclusion our analysis has revealed a large set of new virally-derived genes in Hymenoptera genomes. Those genes were deriving from viruses with any genomic structures, although dsDNA viruses were disproportionately involved in endogenization and domestication. Importantly, our analysis revealed that endogenization rate and the absolute number of domestication events involving dsDNA viruses were increased for endoparasitoids compared to other lifestyles. Among the new cases of endogenization and domestication, we uncovered new events revealing common features with previously known cases of viral domestication by endoparasitoids, such as in the Platygastroidea *Platygaster orseoliae*. This is to our knowledge the first case reported in the superfamily Platygastroideae, thus extending the diversity of Hymenoptera concerned by viral domestication. We propose that the higher rate of endogenization and higher number of domestication events in endoparasitoids is a consequence of the extreme selective pressure exerted by the host immune system on endoparasitoids. This extreme selective pressure may select endoparasitoids for retaining a viral machinery that could helps them address virulence factors in their hosts. We expect this process to be widespread among insect species sharing the same lifestyle.

## Materials and methods
### Genome sampling, assembly correction, and assembly quality

A bioinformatic pipeline mixing sequence homology search, phylogeny, genomic environment, and selective pressure analysis was built to search for viral endogenization and domestication events in Hymenoptera genomes. We used 133 genome assemblies in total, of which 101 were available on public repositories (NCBI and BIPPA databases) and 32 were produced by our laboratory (all SRA reads and assemblies available under the NCBI submission ID: SUB11373855). Concerning the last 32 samples, DNA was extracted on single individuals (usually one female) or a mix of individuals when the specimens were too small using Macherey-Nagel extraction kit, the DNA was then used to construct a true seq nano Illumina library at Genotoul platform (Toulouse, France). The sequences were generated from HiSeq 2500 or HiSeq 3000 machines (15 Gb/sample). The paired-end reads were then quality trimmed using fastqmcf (-q15 –qual-mean 30 -D150, GitHub) and assembled using IDBA-UD (*Peng et al., 2012*). All sample information can be found in *Supplementary file 8* and are available under the NCBI Biosample number: SUB11338872.

The size of the 133 assemblies ranged from 106.14 mb to 2102.30 mb. We kept only genome assemblies containing at least 70% non-missing BUSCO genes (124/133 genomes, *Simão et al., 2015*) (all genome information can be found in *Supplementary file 2*). In addition, when the raw reads were available, we used the MEC pipeline (*Wu et al., 2018*) to correct possible assembly errors. Although some genomes were highly fragmented (such as the 32 genomes we generated since they were obtained using short reads only), the N50 values (min: 3542 bp) were equal to or larger than the expected sizes of genes known to be endogenized and domesticated (average size = 1244,4 bp (sd = 1105 bp)) indicating that most of the putative EVEs should be detected entirely.

Out of the 32 samples sequenced by our laboratory for this study, one (corresponding to *Platygaster orseoliae*) gave unexpected results. After assembly and BUSCO analysis, two sets of contigs were identified: one with only 4x coverage on average, and one with 33x on average. The phylogeny of these different BUSCOs gene sets showed that the low-coverage scaffolds likely belong to an early

diverging lineage of Chalcidoidea (*Figure 1*), whereas the 33x scaffolds belong to the target species *P. orseoliae*. This result suggests that the pool of 10 individuals used for sequencing was likely a mix of two species. A phylogenetic study based on Ultra Conserved Elements (UCEs) obtained from several species of Chalcidoidea (*Rasplus et al., 2020*; *Cruaud et al., 2019*) allowed us to identify the unknown species as a sister to *Aprostocetus sp* (Eulophidae). These UCEs (along with 400 bp of flanking regions) were extracted from the low coverage scaffolds with a custom script. We used a two-step process to assign the unknown sample to species. First, UCEs+ flanking regions were analyzed with a set of UCEs+ flanking regions were obtained from early diverging families of Chalcidoidea by *Cruaud et al., 2019*; *Rasplus et al., 2020* to assign the unknown sample to a family. Then, unknown sequences were analyzed with a larger set of species belonging to the identified family (Eulophidae; loci taken from *Rasplus et al., 2020*). In both cases, only loci that had a sequence for at least 75% of the samples included in the analysis were retained. Loci were aligned with MAFFT (-linsi option; *Katoh and Standley, 2013*). Positions with >90% gaps and sequences with >25% gaps were removed from the alignments using SEQTOOLS (package PASTA; *Mirarab et al., 2015*). The concatenation of all loci was analyzed with IQ-TREE v 2.0.6 (*Minh et al., 2020*) without partitioning. Best models were selected with the Bayesian Information Criterion (BIC) as implemented in ModelFinder (*Kalyaanamoorthy, 2017*). FreeRate models with up to ten categories of rates were included in tests. The candidate tree set for all tree searches was composed of 98 parsimony trees +1 BIONJ tree and only the 20 best initial trees were retained for the NNI search. Statistical support of nodes was assessed with ultrafast bootstrap (UFBoot) (*Minh et al., 2013*) with a minimum correlation coefficient set to 0.99 and 1000 replicates of SH-aLRT tests (*Guindon et al., 2010*). Results of the phylogenetic analyses are presented in *Figure 1—figure supplement 6*. Placement of the unknown species in trees shows that samples of *P. orseoliae* were likely mixed up with a species belonging to the genus *Aprostocetus* (Eulophidae, Tetrastichinae). Given its small size, color, and abundance (265 species described just in Europe), it seems plausible that one specimen of *Aprostocetus* sp. remained unnoticed in the pool of *P. orseoliae* (see phylogeny in *Figure 1—figure supplement 6*). In the figures and tables, the name putative_ *Aprostocetus_*sp was consequently assigned to the unknown sample. However, since the lifestyle and identity of this species are uncertain, we did not include the corresponding scaffolds in the main analysis. The scaffolds belonging to this putative_*Aprostocetus_*sp. (i.e. all scaffolds with a mean coverage <10 X) were removed from the *P. orseoliae* assembly file hosted in NCBI.

## Pipeline outline

EVEs were identified from the 124 Hymenoptera assemblies using a sequence-homology approach against a comprehensive viral protein database (including all categories of viruses: ssDNA, dsDNA, dsRNA, and ssRNA). In order to validate viral endogenization within Hymenoptera genomes, we developed an 'endogenization confidence index' ranging from A to X (*Figure 1—figure supplements 2–7*). This index takes into consideration the presence of eukaryotic genes and/or transposable elements around candidate loci, and scaffolds coverage information (coverage for a valid candidate should be similar to that found in BUSCO containing scaffolds). Finally, the pipeline also included an assessment of the evolutionary history and of the selective regime shaping the candidates (based on $dN/dS$ and/or expression data).

## Hymenoptera phylogeny

The phylogenetic reconstruction of the 124 Hymenoptera species was performed based on a concatenation of the 375 BUSCO proteins. The analysis was conducted by maximum likelihood via Iqtree2 (*Minh et al., 2020*) selecting the best model (*Kalyaanamoorthy et al., 2017*). The tree was rooted via two species of the Coleoptera order (*Anoplophora glabripennis* and *Tribolium castaneum*). Bootstrap scores were evaluated using the UFboot approach (*Hoang et al., 2018*). The results found were consistent with a previous, more comprehensive study (*Peters et al., 2017*).

## Search for viral homology

We collected all protein sequences available in the NCBI virus database (*Hatcher et al., 2017*), removing phage and polydnavirus (virulence genes from wasp origin found within PDVs) sequences. This database contained 849,970 viral protein sequences (download date: 10/10/2019), to which the 40 putative viral proteins encoded by the *Hyposoter didymator* genome were added (so-called

IVSPER sequences, *Volkoff et al., 2010*). The sequence homology search was performed with a BlastX equivalent implemented in Mmseqs2 (*Steinegger and Söding, 2017*) using each genome assembly as queries and the viral proteins collected as a database. The result gave a total of 81,953,678 viral hits (max E-value 5e-04 with an average of 660,916 hits per genome). We kept only candidates with a percentage coverage of the viral protein ≥ 30%, an identity score ≥ 20% and an E-value score <5e-04 (*Figure 1—figure supplement 2*). The threshold parameters were optimized to maximize the detection of the 13 endogenous viral sequences within the genus *Leptopilina* (*Di Giovanni et al., 2020*). Once all the viral hits were recovered, we formed putative EVEs loci (n=238,108) corresponding to the overlap of several viral hits on the same scaffold using the GenomicRanges R package (*Lawrence et al., 2013*; *Figure 1—figure supplement 2*). To remove false positives corresponding to eukaryotic genes rather than viral genes, we then performed another generalist sequence homology search against the Nr database (downloaded the 09/11/20) using mmseqs2 search (-s 7.5, E-value max = 0.0001) (*Figure 1—figure supplements 2–3*). We did not select our candidate based on the best hit, since it does not necessarily reflect the true phylogenetic proximity. Instead, candidates with more than 25 hits with either eukaryotic non-hymenoptera species or prokaryotic species were removed, except if they also had hits with at least 10 different virus species (bits ≥ 50). We chose to eliminate Hymenoptera hits from the database because if a real endogenization event concerns both one of the 124 species of our dataset and some species in the NCBI database, then an apparent 'Hymenoptera' hit will be detected, possibly leading to its (unfair) elimination. Since viral diversity is poorly known, we also kept sequences with even one single viral hit, as long as it did not have more than 5 eukaryotic or prokaryotic hits. Using these filtering criteria we removed a total of 234,036/238,108 (98,3%) candidate loci leaving 4,072 candidates with convincing homology to viral proteins. Note that among these loci a certain proportion actually corresponded to non-endogenized 'free-living' viruses. To study the evolutionary history of these candidate EVEs, we then performed a general protein clustering of all the candidates and the NCBI viral proteins (*Figure 1—figure supplements 2–4*, Mmseqs cluster; thresholds: E-value 0.0001, cov% 30, options: –cluster-mode 1 –cov-mode 0 –cluster-reassign –single-step-clustering *Steinegger and Söding, 2018*).

We eliminated from the dataset the chuviral glycoproteins that have been captured by LTR retrotransposons (*Dezordi et al., 2020*), as these loci have complex histories mostly linked to the transposition activity after endogenization. For this purpose, we systematically searched among the candidates for the presence of TEs within or overlapping with the EVE (more details in *Supplementary file 9*). Only one cluster (Cluster4185) was concerned by such a situation (chuviral glycoproteins overlapping to Gypsy/LTRs). It was detected in 89/124 species (1074 total copies, median = 7 copies/species, max = 244, min = 1), and was probably similar to the one described in *Li et al., 2015*.

## Evolutionary history and selection pressure acting on endogenous loci

### Arguments for endogenization

Among all the candidates for endogenization there were probably false positives that corresponded either to natural contaminants (infecting viruses sequenced at the same time as the eukaryotic genome) or laboratory contaminants (virus accidentally added to the samples). One way to filter these cases was to study (i) the genomic environment (are there other eukaryotic genes or transposable elements on the same scaffolds?) and (ii) metrics such as G+C% (used only for read coverage/GC plots) and scaffold coverage depth around candidate loci (are they the same as scaffolds containing housekeeping genes?). All of these data were used to establish confidence in the endogenization hypothesis, scaled from A to X (*Figure 1—figure supplements 2–7*).

### Scaffolds sequencing depth (*Figure 1—figure supplements 2–5*):

In order to support the hypothesis that a scaffold containing candidate EVEs was part of the Hymenoptera genome, we studied the sequencing depth of the scaffolds. If the sequencing depth of a candidate scaffold was not different from the depth observed in scaffolds containing BUSCO genes, then this scaffold was likely endogenized into the Hymenoptera genome. Hence, when DNA reads were available (*Figure 1—figure supplement 7*), we measured this metric by mapping the reads on the assemblies using hisat2 v 2.2.0 (*Kim et al., 2019*). An empirical p-value was then calculated for each scaffold containing a candidate EVE. To calculate this empirical p-value, we sampled 500 loci of the size of the scaffold of interest within BUSCO scaffolds. These 500 samples represented a null

distribution for a scaffold belonging to the Hymenoptera genome. The p-value then corresponded to the proportion of BUSCO depth values that were more extreme than the one observed in the candidate scaffold (two-sided test). We used a threshold of 5% and a 5% FDR (multiply python package *Puoliväli et al., 2020*).

## Genomic environment and scaffold size (*Figure 1—figure supplements 2–6*):

Another way to rule out contaminating scaffolds were to look for the presence of eukaryotic genes and transposable elements in the scaffolds containing candidate EVEs, assuming that their presence in a viral scaffold is unlikely. Indeed, so far, very few viral genomes have been shown to contain transposable elements (*Miller and Miller, 1982*; *Gilbert et al., 2014*; *Gilbert et al., 2016*; *Gilbert and Cordaux, 2017*; *Loiseau et al., 2020*) because TE insertions are mostly deleterious and are, therefore, quickly eliminated by negative selection (*Gilbert et al., 2016*; *Gilbert and Cordaux, 2017*). We searched for transposable elements with a BlastX-like approach (implemented in Mmseqs2 search -s 7.5), taking as query the scaffolds of interest and as database the protein sequences of the transposable element (TE) available in RepeatModeler database (RepeatPeps, v2.0.2) (*Flynn et al., 2020*). We only kept hits with an E-value <1e-10 and with a query alignment greater than 100 amino acids. We then merged all overlapping hits and counted the number of TEs for each scaffold. To find eukaryotic genes within genomes we used Augustus v3.3.3 (*Stanke et al., 2004*) with BUSCO training and then assigned a taxonomy to these genes via sequence homology with Uniprot/Swissprot database using mmseqs2 search (*Steinegger and Söding, 2017*), and only retained genes assigned to insects. Accordingly, the scaffolds were scored as follows (*Figure 1—figure supplements 2–7*):

- A: scaffolds with a corrected coverage p-value >0.05 and at least one eukaryotic gene and/or one repeat element,
- B: scaffolds with at least one eukaryotic gene and/or one repeat element but no coverage data available,
- C: scaffolds with a corrected coverage p-value >0.05 and neither eukaryotic gene nor transposable element,
- D: scaffolds with a corrected coverage p-value <0.05 and whose coverage value was higher than the average of the scaffolds containing BUSCOs (as it is difficult to imagine that an endogenized scaffold presents a lower coverage than expected, whereas a higher coverage could correspond to the presence of repeated elements that inflate the coverage of the scaffold for example) but with at least five eukaryotic genes and/or a repeated element (in total),
- E: scaffolds presenting no DNA seq coverage data available and no eukaryotic gene nor transposable element detected,
- F: scaffolds presenting a corrected p-value of coverage <0.05 and less than five eukaryotic genes without any transposable elements; this category may rather correspond to free-living viruses.
- X: scaffolds with a corrected p-value <0.05 and neither eukaryotic gene nor transposable element; This category may rather correspond to free-living viruses.

Only scaffolds scored as A, B, C, or D were considered as endogenized, whereas E scaffolds were not sufficiently supported by the data to be considered as endogenized. They were thus discarded from the main analysis. Scaffolds scored as F and X were rather considered as free-living viruses.

## Inference of endogenization events

Because several EVEs may derive from the same endogenization event, we sought to aggregate EVEs into unique events. We aggregated into a single event, firstly (i) all the EVEs present on the same scaffolds, and secondly (ii) all the EVEs that presented the same taxonomic assignment at the level of the viral family. These two steps were sufficient to aggregate EVEs in the simplest case of events involving only one species (but possibly several EVEs).

To further characterize the endogenization events including more than one Hymenoptera species, we also relied on phylogenetic inference. To this end, the protein sequences belonging to each of the clusters (containing both viral proteins and candidate EVEs) were first aligned with clustalo v1.2.4 (*Sievers et al., 2011*) in order to merge possible candidate loci (which may in fact correspond to various HSPs). All loci (=HSPs) within the same scaffold presenting no overlap in the alignment

were thus merged, as they probably correspond to multiple HSPs and are not duplications. We then performed a new codon alignment from the augmented sequences in the clusters using the MACSE v2 alignsequence program (*Ranwez et al., 2018*; *Figure 1—figure supplements 2–8*). This alignment allowed us to obtain a protein and nucleotide codon alignment. We used the protein alignment to infer the phylogeny of each cluster with the program Iqtree2 v2.1.2 (*Minh et al., 2020*) (-m MFP -alrt 1000 (partitioned))(*Figure 1—figure supplements 2–9*). No trimming was performed at the amino-acid level, since this may result in a loss of topology information (*Tan et al., 2015*; *Ranwez and Chantret, 2020*). However, since it can affect branch length, only codon alignment was trimmed at the protein level via Trimal v1.2 (*Figure 1—figure supplement 2*) (-backtrans -automated1) (*Capella-Gutiérrez et al., 2009*). We then exploited the information from the cluster phylogenies to form the endogenization events. EVEs potentially deriving from the same event should be supported by the formation of the same well-supported monophyletic clade (bootstrap score >80) both in the gene tree and the Hymenoptera tree (allowing gene losses in 20% of the species concerned by the monophyletic group). EVEs were possibly aggregated within the same event only if the Hymenoptera belonged to the same family. (*Figure 1—figure supplement 2*). Finally, the clustering of multiple EVEs within the same scaffold in one species was used to aggregate the homologous EVEs found in a related species within the same shared event, even if they were on different scaffolds (*Figure 1—figure supplement 2*). For details, see some canonical examples in *Figure 1—figure supplement 3*.

The majority of EVEs were found within scaffolds considered as endogenized (A-D score) (*Figure 1—figure supplement 8B*). Besides, the distribution of EVEs among scores was roughly identical for each viral genomic structures (*Figure 1—figure supplement 8*). The main difference was in the fraction of EVEs annotated as X or F. This fraction was higher for DNA viruses compared to RNA viruses (*Figure 1—figure supplement 8B*). Because these scaffolds likely belong to free-living viruses that have been sequenced together with the insect DNA, it is not a surprise to observe an excess of DNA viruses, since DNA and not RNA were sequenced.

For events shared by several species, we were also able to analyze gene synteny around putative EVEs. To do this, we conducted the equivalent of an all vs all TblastX (Mmseqs2 search –search-type 4, max E-value=1e-07) between all the candidate loci within a putative event (deduced from the phylogenetic inference), and then looked for hits (HSPs) between homologous EVEs around the insertions. Because it is possible to find homology between two genomic regions that does not correspond to orthology, for example, because of the presence of conserved domains, we had to define a threshold to identify with confidence the orthology signal. We, therefore, conducted simulations to define this value, based on the well-assembled genome of *Cotesia congregata* (GCA_905319865.3) by simply performing the same all vs all blast analysis against itself (as if the two species considered had the same genome). Based on this, we defined two types of simulated EVEs, (i) independently endogenized EVEs in the genomes of the two 'species'. This is simply simulated by randomly selecting two different regions in the genomes, and (ii) a shared simulated EVE that was acquired by their common 'ancestor'. This is simulated by selecting the same random genomic location in both 'genomes'. We then counted the total length of the HSPs found around the simulated insertions all along the corresponding scaffold (i and ii). As the result will obviously depend on scaffold length, we performed these simulations on several scaffold lengths (100000000 bp, 100,00,000 bp, 100,00,00 bp, 100,000 bp and 10,000 bp). We conducted 500 simulations in each scenario, and we measured the cumulative length of homologous sequences by counting the sum of HSPs (bit score >50). We then defined a threshold for each windows size in order to minimize for the false-positive (FP) and maximize true-positifs (TP) (thresholds 100,000,000 bp = 172737 bp (FP = 0.012, TP = 0.922); 100,00,000 bp = 74262 bp (FP = 0.012, TP = 0.878); 100,00,00 bp = 21000 bp (FP = 0.014, TP = 0.28); 100,000 bp = 1332 bp (FP = 0.012 TP = 0.198) and 10,000 bp = 180 bp (FP = 0.008, TP = 0.208)).

Events were linked to viral families based on the closest match information between the viral blastx (GenBank accession number and/or viral protein and/or viral species) and the classification proposed in *Shi et al., 2016*.

## Arguments for domestication

One way to test for the domestication of an EVEs (dEVEs) was to estimate the ratio (omega) of the number of nonsynonymous substitutions per non-synonymous site (*dN*), to the number of synonymous substitutions per synonymous site (*dS*). If EVEs are evolving neutrally, then the ratio is expected

to be equal to 1, whereas if the EVE is under purifying selection, *dN/dS* is expected to be lower than 1. We conducted this analysis on trimmed codon alignments from (*Figure 1—figure supplement 2*) via the codeml algorithm from PAML *Yang, 2007* used through the ETE3 package (*Huerta-Cepas et al., 2016*) (model *Muse and Gaut, 1994*). We then used a branch model to test the deviation from the null model in which marked branches (called foreground) wich corresponded to the monophyletic EVE clade evolved under a neutral scenario ($\chi^2$ test). The *dN/dS* estimated for the whole clade is then the average of each branch of the clade. The p-values were then adjusted by selecting an FDR of 0.05 (*Puoliväli et al., 2020*), and we estimated the standard errors of *dN/dS* that maximized the likelihood (option getSE = 1). *dN/dS* with dS greater than 10 were removed, since this indicates substitution saturation (*Figure 1—figure supplement 2*).

The other way we choose to study the domesticated nature of a viral gene was to study their expression profile (*Figure 1—figure supplement 2*). We reasoned that domesticated genes are likely to be significantly expressed. To test this, when RNAseq reads were available on NCBI (SRA), we mapped them on the assembled genomes (until reaching 300x coverage as far as possible). Using the TPMCalculator program (*Vera Alvarez et al., 2019*), we measured expression in ovaries and the whole body if available or alternatively in any tissue (see *Supplementary file 10*). An EVE was considered as domesticated if the gene was expressed with a Transcripts Per Kilobase Million (TPM) index above 1000. This threshold was chosen based on the median value observed for control EVEs (718.70 TPM), rounded up to 1000 TPM to be conservative. We measured the accuracy of this metric using EVEs for which both TPM and *dN/dS* calculations were possible: among the 36 genes having a TPM >1000, 33 also had a *dN/dS* significantly below 1 suggesting that inferring domestication based on TPM >1000 was consistent with *dN/dS* test with a 0.9166 probability. Finally, based on the idea that an active EVE should encode a protein with a similar length to the donor virus, we calculated the actual viral protein sequence length using the orfipy algorithm (*Singh and Wurtele, 2021*; *Figure 1—figure supplement 2*).

A possible bias when comparing the effect of lifestyles on domesticated elements could come from a difference of RNAseq reads availability depending on the lifestyle, which may result in a different number of EVEs considered as domesticated. A GLM binomial analysis did not reveal any correlation between RNAseq data availability and lifestyle (endoparasitoid = Slope(SE)=0.21 (0.62), p=0.73; free-living=Slope(SE)=0.40 (0.57), p=0.49 using ectoparasitoid as intercept).

## Sensitivity and specificity of the analysis

### Capacity to find EVEs

Among the species included in our dataset, seven were known to contain a domesticated virus (two with similar PDV (*Bézier et al., 2009*), and five with different VLPs (*Pichon et al., 2015*; *Burke, 2019*; *Di Giovanni et al., 2020*), corresponding to four independent endogenization events). Our pipeline was able to detect the vast majority of the corresponding virally-derived genes (88.6%, details in *Supplementary file 3* and *Figure 1—figure supplement 9*). The 11.14% false negatives corresponded to sequences that were too divergent or with a region of similarity is too small to be detected by our pipeline. We found that 88.7% of the control EVEs were located within that scaffolds scored as A (i.e. having a depth of coverage falling within the distribution of those containing BUSCO genes, as well as having one or more eukaryotic genes and/or transposable elements in the vicinity). Since the remaining 11.3% were scored either C (7.64%) or D (3.66%) (*Supplementary file 3*), we considered candidates within the range A-D as valid candidates for endogenization. On the contrary, scaffolds annotated as F or X were rather considered as free-living viruses since they did not show eukaryotic genes or TE in their vicinity and had different coverage compared to BUSCO-containing scaffolds. Scaffolds classified as E were of unclear status and discarded. Capacity to find domesticated EVEs (dEVEs) An EVE was considered as domesticated if the *dN/dS* ratio was significantly below 1 or if TPM was above 1000. When *dN/dS* computations were possible (for 75/152 control EVEs), our pipeline considered the control EVEs as being under purifying selection in 70.39% of the cases. Overall, by combining the two metrics (*dN/dS* and TPM), our pipeline identified 69.04% of the control locus as being domesticated (*Supplementary file 3*). Capacity to infer events of endogenization (EVEs events) Among the control species, the pipeline correctly inferred the expected four independent events: (1) *Leptopilina boulardi/Leptopilina clavipes/Leptopilina heterotoma* (*Di Giovanni et al., 2020*), (2) *Venturia canescens* (*Pichon et al., 2015*), (3) *Fopius arisanus* (*Burke et al., 2018*), and (4) *Cotesia*

*vestalis*/*Microplitis demolitor* (*Bézier et al., 2009*; *Supplementary file 3*). However, in addition to the expected unique shared event concerning the *M. demolitor* and *C. vestalis* species, our pipeline inferred two additional events, each specific to one lineage. This was due to the fact that two genes were not detected by our pipeline as shared by *M. demolitor* and *C. vestalis*, either because they are effectively not shared (for three of them: HzNVorf118, like-*pif-4* (*19* kda), *fen-1*), or because of some false negative in one of the two lineages (for one of them:*p33* (*ac92*)). For a canonical examples in our dataset result, please see (*Figure 1—figure supplement 3*).

## Assessing the distribution of virus infecting insects

We estimated the number of viral species infecting insect species based on the virushostdb database (version of 24/03/2023 on *Mihara et al., 2016*) which lists a wide diversity of viral species associated with their putative hosts. We kept only viruses found in interaction with insects. Genomic structures were retrieved through the ICTV report (V2022_MSL38) and information available in ViralZone (all viral species details can be found in *Supplementary file 5*). We counted the number of viruses per genomic structure, and viruses from unknown genomic structures were discarded. Importantly, the distribution of viruses among genomic structures found in Hymenoptera was not different from the distribution found in the whole Insecta class ($chi^2$ = 6.39, d.f.=3, p-value = 0.094).

To study the distribution of negative versus positive-stranded viruses among RNA viruses infecting insects, we also relied on virushostdb complemented by two important exploratory studies focusing on RNA viruses (*Shi et al., 2016*; *Wu et al., 2020*). In total, 2609 viral species infecting insects were considered (detail: ssRNA(-) = 597 sp, ssRNA(+) = 1240 sp, ssDNA = 78 sp, dsRNA = 401 sp, dsDNA = 145 sp, Unknown = 148 sp). The Partiti-Picobirna, Narna-Levi, Mono-Chu, Bunya-Arenao, Luteo-Sobemo, Hepe-Virga, and Picorna-Calici clades correspond to viral clades proposed by *Shi et al., 2016*.

## *Naldaviricetes* phylogenetic inference

In order to infer the phylogeny of *Naldaviricetes*, we retrieved all predicted ORFs from a set of 25 dsDNA viruses including *Baculoviridae*, *Nudiviridae*, *Hytrosaviridae*, the LbFV-like and AmFV-like viruses, and *Nimaviridae* were used as outgroup. We then merged these ORFs with all the candidate EVEs extracted from Hymenoptera genomes and performed an all versus all blastp analysis using Mmseqs2 (*Steinegger and Söding, 2017*). All sequences sharing at least an alignment bit score >50 and a coverage >15% were then clustered together. We kept only clusters with more than three species (n=142 clusters). Each cluster (or partition) were then aligned using clustalo v1.2.4 (*Sievers et al., 2011*) and trimmed using Trimal v1.2 (-automated1) (*Capella-Gutiérrez et al., 2009*). Each partition was then concatenated to form one unique protein alignment using the catfasta2phyml. pl script. The *Naldaviricetes* phylogeny was then inferred using Iqtree2 (*Minh et al., 2020*) using the best model for each partition (*Kalyaanamoorthy et al., 2017*) (-m MFP -alrt 1000 -bb 1000).

## Divergence time estimation

We time-calibrated the inferred phylogenetic tree using a Bayesian approach on RevBayes v1.1.1 (*Höhna et al., 2014*) and information on five fossils selected by *Peters et al., 2017*. Reduction of the supermatrix became necessary to overcome computational limitations when estimating node ages resulting from the large size of the concatenated BUSCO supermatrix (nsites = 228,009). We then generated one fasta file with a random draw without the replacement of 20,000 sites from the supermatrix. Evaluation of the phylogenetic likelihood being the most expensive operation when calculating the posterior density, we decided to use the method developed in *Szöllősi et al., 2022* to reduce computational cost and approximate the phylogenetic likelihood using a two-step approach. In the first step, the posterior distribution of branch lengths measured in the expected number of substitutions is obtained for the fixed unrooted topology of using a standard MCMC analysis (100,000 iterations, three chains, 5000 burn-in, tuningInterval = 200). The obtained posterior distribution is then used to calculate the posterior mean and posterior variance of the branch length for each branch of the unrooted topology. In the second step, we date the phylogeny using a relaxed clock model and calibrations (500,000 iterations, four chains, 5000 burn-in, tuningInterval = 200). Calibration of the root was done using a uniform law between 300 and 412 Mya. To verify that MCMC analyses converged to the same posterior distribution, for both steps we computed the effective sample size

and applied the Kolmogorov-Smirnov test using the package convenience v1.0.0 with a minimum ESS threshold of 100 (however, due to an excessive demand for resources, we were unable to achieve the sampling value of an ESS of a minimum of 100 for 46/389 parameters (min = 44.25)).

## Ancestral state reconstruction

To explore the dynamics of EVEs gain in relation to lifestyle, we first had to reconstruct the ancestral lifestyle states of the Hymenoptera used in this study. This was achieved using a Bayesian approach implemented in RevBayes v1.1.1 (*Höhna et al., 2014*). The lifestyles of the Hymenoptera species used in this study were deduced from various sources (details and sources in *Supplementary file 2*). Since lifestyle characters are probably not equally likely to change from any one state to any other state, we choose the Mk model with relaxed settings allowing unequal transition rates. Thus, we assumed six different rates with an exponential prior distribution. Before running the MCMC chains, we made a preliminary MCMC simulation used to auto-tune the moves to improve the mixing of the MCMC analysis with 1000 generations and a tuning interval of 300. We then ran two independent MCMC analyses, each set to run for 200,000 cycles, sampling every 200 cycles, and discarding the first 50,000 cycles as burn-in. To verify that MCMC analyses converged to the same posterior distribution, we computed the effective sample size and applied the KolmogoRov-Smirnov test using the package convenience v1.0.0 with a minimum ESS threshold of 100. The MCMC chain was subsampled to provide 1000 samples. At each sample, ancestral states were reconstructed for all nodes of the phylogeny. We assumed that the state assigned to a node was constant throughout the branch leading to that node.

## Test of the lifestyle effect on viral endogenization and domestication

In order to test the lifestyle effect on the propensity to integrate and domesticate viral elements, we first randomly sampled 1000 probable ancestral state scenarios to take into account the uncertainty in the estimates of the ancestral states of the nodes. Because a lot of branches had no EVE endogenization inferred, we ran zero-inflated negative-binomial GLM model, for each of these 1000 scenarios such that (GLM(Number EVEs ~free-living + endoparasitoid + ectoparasitoid * Branch_length, family = zero-inflated neg binomial)). We eliminated all branches older than 160 million years because they are too old for our analysis to detect events (the oldest event detected by our analysis is around 140 mya) that could artificially inflates the zero count. The model was implemented in stan language using the R package brms (seed = 12345,, thin = 5, nchains = 4, niter = 10000) (*Bürkner, 2017*; *Bürkner, 2018*). The same analysis was carried out by splitting the free-living category into two subcategories, namely eusocial and free-living. A new GLM model was then built (GLM(Number EVEs ~free-living+eusocial + endoparasitoid + ectoparasitoid * Branch_length, family = zero-inflated neg binomial)). Posterior predictive check was done using the package brmsfit in order to check that the model was correctly predicting the proportion of zeros. Indices relevant to describe and characterize the posterior distributions were computed using the R package BayestestR (*Makowski et al., 2010*). Autocorrelation was studied using the effective sample size index (ESS) with a value greater than 1000 being sufficient for stable estimates (*Bürkner, 2017*). The convergence of Markov chains was evaluated by a Rhat statistic equal to 1. All the posterior coefficient estimated values were then pooled together (after checking the convergence of all chains via the GelmanRubin function in R *Bolstad, 2009*) and compared between the free-living, endoparasitoid and ectoparasitoid modalities.

To calculate the rate of domestication independent of the rate of endogenization, we built a binomial logistic regression model in a Bayesian framework, specifying the number of domesticated EVEs (or Events) (the numerator) relative to the total number of EVEs or Events inferred by our pipeline (the denominator). These binomial models allowed us to test whether the probability of domestication after endogenization correlated with lifestyle by controlling for the endogenization input (the denominator). Thus, for each of the 1000 lifestyle scenarios, we ran a binomial brms model with a logit link such that brms(Number dEVEs/dEvents | trials(Number EVEs/Events)~lifestyle + Branch length).

Before analyzing the data, we checked that the inferences did not depend on the quality of the genomes selected for analysis. We found a significant effect of the lifestyle on the N50 and percentage of complete + partial BUSCO in the assemblies (Kruskal-Wallis rank sum test p-values, respectively = 3.192e-10 and 1.26e-14). Furthermore, a pairwise Wilcoxon test with p-values adjusted with the Bonferroni method revealed a significantly higher values of N50 and %complete + partial BUSCO in

genome assemblies from free-living species compared to endo and ectoparasitoids species (p-value <0.05). The same test using the total assembly length in bp did not reveal any difference between the three lifestyles (p-value >0.05). Overall, free-living species have better assemblies. Because better assembly quality should facilitate the discovery of endogenous viral elements both by sequence homology detection and by a better assessment of the endogenized nature of the EVE (scaffolds A, B, C, and D), we should thus underestimate the number of EVEs in endo and ectoparasitoid species compared to free-living species. Since our analysis led to the opposite conclusion, our results cannot be explained by this feature of the dataset.

In our main analysis, we only considered EVEs with scores from A to D as confidently endogenized. To test the impact of this scoring system on our conclusions, we ran the very same analysis on either top-confidence EVEs (A score) (*Figure 4—figure supplement 1*), or on a more relaxed dataset also including poorly-scored EVEs (A-F). The main conclusions of the manuscript, in particular regarding the link between endoparasitoidism and dsDNA virus endogenization and domestication, were consistently reached (see all statistical summaries of the Bayesian GLM models in the *Supplementary file 6*).

## Acknowledgements

This work was performed using the computing facilities of the CC LBBE/PRABI. We thank Clément Gilbert and Jean-Michel Drezen for helpful discussions, three anonymous reviewers for helpful comments, and Elijah J Talamas, Nicolas Ris, Lene Sigsgaard for providing specimens.

## Additional information

### Funding

| Funder | Grant reference number | Author |
| --- | --- | --- |
| Agence Nationale de la Recherche | 11-JSV7-0011 | Julien Varaldi |
| Agence Nationale de la Recherche | 17-CE02-0021 | Sylvain Charlat |

The funders had no role in study design, data collection and interpretation, or the decision to submit the work for publication.

### Author contributions

Benjamin Guinet, Conceptualization, Data curation, Formal analysis, Methodology, Writing – original draft, Writing – review and editing; David Lepetit, Data curation; Sylvain Charlat, Resources, Writing – review and editing; Peter N Buhl, David G Notton, Jean-Yves Rasplus, Julia Stigenberg, Resources; Astrid Cruaud, Resources, Formal analysis; Damien M de Vienne, Bastien Boussau, Methodology, Writing – review and editing; Julien Varaldi, Conceptualization, Supervision, Funding acquisition, Writing – original draft, Project administration, Writing – review and editing

### Author ORCIDs

Benjamin Guinet http://orcid.org/0000-0002-9922-2118
David Lepetit http://orcid.org/0000-0002-8968-1994
Sylvain Charlat http://orcid.org/0000-0003-0760-0087
David G Notton http://orcid.org/0000-0002-8933-7915
Jean-Yves Rasplus http://orcid.org/0000-0001-8614-6665
Damien M de Vienne http://orcid.org/0000-0001-9532-5251
Bastien Boussau http://orcid.org/0000-0003-0776-4460
Julien Varaldi http://orcid.org/0000-0002-2100-1542

### Decision letter and Author response
Decision letter https://doi.org/10.7554/eLife.85993.sa1
Author response https://doi.org/10.7554/eLife.85993.sa2

## Additional files

### Supplementary files

- Supplementary file 1. Summary statistics table for candidate EVEs.

- Supplementary file 2. General information regarding the species used in this study.

- Supplementary file 3. Summary statistics table for control endogenous viral elements (EVEs).

- Supplementary file 4. Information for individual loci. Endogenized loci (scoring from A to D) are displayed in the first sheet, whereas exogenous loci (from E to X) are displayed in the second sheet.

- Supplementary file 5. Table listing the names of virus species known to probably interact with insects. The data is taken from the virushostdb database (*Mihara et al., 2016*) (version of 24/03/2023), which lists a wide variety of virus species associated with their presumed hosts. We have also added two important exploratory studies of RNA viruses (*Shi et al., 2016*; *Wu et al., 2020*). The viral genomic structures associated with the viruses were retrieved from the ICTV website (V2022_MSL38). Each column represents the information retrieved for each virus species from one of the three sources listed. The Hostdb_Host_lineage column corresponds to the information of the insect host observed interacting with the virus of interest. If a column with the suffix 'Shi' or 'Haoming' contains information for a virus species, then this means that this RNA virus species was found in their dataset in an insect.

- Supplementary file 6. Datasets and detailed statistics are presented in the manuscript.

- Supplementary file 7. Additional information from the double-stranded DNA (dsDNA) *Naldaviricetes* phylogenetic analysis in *Figure 5*, including the best partition models chosen for each partition and the number of genes used for each species of the tree.

- Supplementary file 8. Biosample information regarding the 34 Hymenoptera species sequenced for this study.

- Supplementary file 9. Table representing the overlap between transposable elements and the clusters of homologous endogenous viral elements (EVEs). The transposable elements were inferred using the RepeatModeler RepeatPeps database.

- Supplementary file 10. Information on the RNAseq datasets used in this study.

- Supplementary file 11. Details of the tblastn analysis for Platygaster orseoliae endogenous viral elements (PoEFV) and complementary information.

- Supplementary file 12. File including all the cluster phylogenies. Leafs highlighted in green represent endogenous viral elements (EVEs) (scored from A to D), while leafs highlighted in red represent free-living viruses or loci annotated as putative free-living sequences (scored from D to X). The letter at the end of the taxon label represents the endogenization score assigned to the candidate. The assigned viral family of the free-living genes appears next to the pipe. The numbers right next to the 'Event' refers to the assigned Event number. If EVEs were found under selection (either by RNAseq or dN/dS analysis), the end of the leaf name will be 'Selective_pressure_YES,' while if not, the name will be 'Selective_pressure_NO.' Ultra-Fast Bootstrap values can be found next to the nodes of the phylogenies. For each phylogeny, the putative consensus protein name as well as the putative viral family is given at the top of the figure.

- MDAR checklist

### Data availability

All sequencing data are available at NCBI via the BioProject accession number NCBI: PRJNA826991. All pipelines, statistical as well as figures scripts are available under the GitHub repository : [https://github.com/BenjaminGuinet/Viral-domestication-among-Hymenoptera, (copy archived at swh:1:rev:a96c55309b929044f19c0b267523bd31170b14c2)].

The following dataset was generated:

| Author(s) | Year | Dataset title | Dataset URL | Database and Identifier |
|---|---|---|---|---|
| Guinet B, Lepetit D, Charlat S, Buhl PN, Notton DG, Cruaud A, Rasplus J-Y, Stigenberg J, de Vienne DM, Bastien B, Varaldi J | 2023 | Genome assemblies and SRA reads of parasitoid wasps | https://www.ncbi.nlm.nih.gov/bioproject/PRJNA826991 | NCBI BioProject, PRJNA826991 |

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
