## [Editor Report]

This important manuscript employs a rigorous and multi-pronged comparative genomics approach to unravel how lifestyle modulates the acquisition and domestication of viral genetic elements in the genomes of hymenopteran insects. Using an extensive dataset of over 120 hymenopteran genomes, the authors provide convincing evidence that endoparasitism (where parasite development occurs within hosts) facilitates the uptake and domestication of double-stranded DNA viral elements.

---

## [Decision Letter]

**Decision letter after peer review:**

Thank you for submitting your article "Endoparasitoid lifestyle promotes endogenization and domestication of dsDNA viruses" for consideration by *eLife*. Your article has been reviewed by 3 peer reviewers, and the evaluation has been overseen by a Reviewing Editor and Christian Landry as the Senior Editor. The reviewers have opted to remain anonymous.

Essential revisions:

All three reviewers made excellent and very constructive comments that would greatly improve your manuscript. In particular, please address:

1) Lack of inclusion of genomic data from hymenopteran species made available after 2019.

2) Oversimplification of categorizing hymenopterans either as free-living, endoparasitoids, or ectoparasitoids and its impact on the results.

3) Examine the impact of the scoring system for EVE detection (Scaffold endogenization score).

*Reviewer #1 (Recommendations for the authors):*

Please replace the use of "Nb" throughout the manuscript (is it meant to indicate "number").

Line 104: are the authors referring to 13 EVE events or 13 EVE genes here? At this point, it would be good to include another example with EVE events enumerated so that readers can directly relate what's written to the numbers included in Figure 1. The given example is unverifiable without delving into the supplementary information.

Figure 1:

– The numbers listed in white boxes for the number of endogenization events do not seem to match with the number of EVEs plotted in the bar graph to the right. For example, how can Neodiprion leconti have 4 endogenization events but six EVEs from different virus families? Examination of the supplementary information makes it more clear that sometimes EVEs from different virus families are grouped together based on phylogenetic or genomic context. This should be stated early enough in the manuscript that readers don't get confused when trying to interpret Figure 1.

– The labels are very hard to see without using a magnified pdf viewer. I suggest that the journal moves the legend to the next page so that the figure can be printed larger.

– "Campopleginae" should be listed as "Campopleginae sp". As written it suggests that this taxon represents the entire sub-family of wasps.

– The weight of the black line around the pie charts indicating shared endogenization events differs between charts. The line weight appears to have no meaning so would prevent confusion if they were consistent.

Line 118: Do the authors mean to say that 113 underwent more than or at least (>) 1 endogenization event?

Line 126: It is mentioned that 12 endogenization events involved the integration of more than 4 viral genes. From Figure 1, I count 14 (Campopleginae sp, Venturia canescens, Fopius arisanus, Cotesia vestalis, Microplitis demolitor, three Leptopilina species, Platygaster orseoliae, Eurytoma brunniventris, Harpegnathos saltator, Pseudomyrmex gracilis, Aphaenogaster picea, Aprostocetus sp?). Which is correct?

Supplementary file 2 – when this file is opened in excel, the column headings are shifted so that they are not present in the correct column. Please define the headings in a separate tab or document – currently, it is very difficult to interpret the information within. Which column indicates domestication (or not)?

Figure 2:

– In 2A, should an average/median "+" sign be present in the ssRNA column?

– In 2B, should IVSPERs be in orange?

– 2C, the numbers indicated don't make sense based on other results. Are the EVE numbers inclusive of dEVE numbers? Earlier in the results, it was stated that there are 12 events involving >4 genes for dsDNA viruses. Is this figure depicting events or genes? Line 127 implies genes, and the figure legends and titles imply events.

– The reference to the "excel tab" information needs to be more explicit

Line 141 and 144: Is there conflicting information presented here about ssRNA viruses being "under-represented" and then "over-represented"? Where is the data presented to support these statements?

Figure 3: The colour in the legend is different from the colour in the figure for ectoparasitioids,

Figure 5: Where are the methods that were used to make this figure? It is surprising that 142 genes were used to make the alignment. What is the percentage of missing data for each taxon?

Line 242: After describing 139 convincing EVEs from P. orseoliae, then 44/89 are mentioned as presenting signs of domestication. Where does the denominator of 89 come from?

Line 255: Does AmFV have RNA polymerase subunits or just a single gene encoding the RNA polymerase? If the former, which subunit does AmFV_0047 encode?

Line 275: "we will first discuss"… "a hypothesis for" should be inserted here.

Line 510: "transportable elements" needs to be fixed.

*Reviewer #2 (Recommendations for the authors):*

Line 117 – 118 – this sentence is unclear, please revise.

Figure 2 – I would advise adding grid lines to the plots. Now it is very difficult to see how high the bars go for certain lines (I had to use my ruler).

Line 152 – Looking at Figure 3A, it surprises me that ssDNA viruses are not underrepresented in endoparasitoid wasps. The observed percentage is half of the expected percentage.

Line 215 – 216 – is a gene-loss scenario not equally likely?

Figure 5 – what do the question marks exactly mean here?

Line 241 – this was on scaffolds of type A. I would specify here to avoid confusion.

Line 262 – 263 – I am not sure how an absence of raw reads is possible.

Line 312 "at early times" please clarify.

Line 331 – 336 – I would place this section in supplemental.

Line 341 – 352 – This section recapitulates to a large extent content that was already outlined in the introduction. For sake of brevity, I would remove it.

*Reviewer #3 (Recommendations for the authors):*

In line with my public comment on the scoring system, I feel a more thorough investigation of its impact on the results and conclusions should be conducted such as:

– Repeating key conclusions on endogenization/domestication bias for example in being over-inclusive (A to F).

– Giving a fair representation of the score distribution among different virus genome type.

– Sharing complete S2 Table (not truncated to the D score).

This would give a better representation of the results. Being over-inclusive has its risks here, and I would be more trusting of the authors' original results, but it might also explain some of the bias (maybe more ssRNA-derived EVE are low scoring, which could explain the bias observed?)

---

## [Author Response]

Reviewer #1 (Recommendations for the authors):Please replace the use of "Nb" throughout the manuscript (is it meant to indicate "number").

We changed that.

Line 104: are the authors referring to 13 EVE events or 13 EVE genes here? At this point, it would be good to include another example with EVE events enumerated so that readers can directly relate what's written to the numbers included in Figure 1. The given example is unverifiable without delving into the supplementary information.

We are referring to 13 EVEs that concern one single Event. To further clarify this, we added a few sentences in this paragraph:

Lines 106-114

“As an example, let's consider the single endogenization event involving 13 EVEs that occurred in the common ancestor of *Leptopilina* species (Digiovanni et al., 2020). In this wasp genus, based on previous findings, we expect the 39 EVEs to be grouped into a single endogenization event. Our pipeline appropriately detected 36 EVEs (out of 39) and correctly aggregated them into a single endogenization event mapped on the branch leading to the *Leptopilina* genus. Thus, in Figure1 we can observe a pie chart at the node corresponding to the common ancestor of the three *Leptopilina* species *L.boulardi*, *L.clavipes* and *L.heterotoma*. Most of this pie chart is blue, which corresponds to the putative donor viral family, i.e., LbFV-like, and is surrounded by a black border, indicating that the genes involved are inferred as domesticated. The number of EVEs and dEVEs (n=12/13) for each of the 3 species is then plotted along the horizontal barplots with the same colour code (see Figure1 and Figure 1—figure supplement 3 for more canonical examples).” Additionally, another example is provided in the legend of Figure1.

Figure 1:– The numbers listed in white boxes for the number of endogenization events do not seem to match with the number of EVEs plotted in the bar graph to the right. For example, how can Neodiprion leconti have 4 endogenization events but six EVEs from different virus families? Examination of the supplementary information makes it more clear that sometimes EVEs from different virus families are grouped together based on phylogenetic or genomic context. This should be stated early enough in the manuscript that readers don't get confused when trying to interpret Figure 1.

To clarify, we added a sentence at the very beginning of the result paragraph: “Since several EVEs may enter into the genome during a single endogenization event, we grouped into the same event EVEs that were localized in the same scaffold (only for viruses having similar genomic structure), and/or that derived from the same putative viral family.”. (lines 95-97)

– The labels are very hard to see without using a magnified pdf viewer. I suggest that the journal moves the legend to the next page so that the figure can be printed larger.

We agree and we also increased the label size.

– "Campopleginae" should be listed as "Campopleginae sp". As written it suggests that this taxon represents the entire sub-family of wasps.

We have changed it.

– The weight of the black line around the pie charts indicating shared endogenization events differs between charts. The line weight appears to have no meaning so would prevent confusion if they were consistent.

The weight of the black line was proportional to the size pie chart, but we changed it to keep it constant whatever the size of the pie chart.

Line 118: Do the authors mean to say that 113 underwent more than or at least (>) 1 endogenization event?

We meant at least. To clarify, we modified the text as follow: “Among the 124 species under study, 113 underwent at least one endogenization event, with a maximum of 14 events and a median of 3 (Figure1). (Lines 122-124).

Line 126: It is mentioned that 12 endogenization events involved the integration of more than 4 viral genes. From Figure 1, I count 14 (Campopleginae sp, Venturia canescens, Fopius arisanus, Cotesia vestalis, Microplitis demolitor, three Leptopilina species, Platygaster orseoliae, Eurytoma brunniventris, Harpegnathos saltator, Pseudomyrmex gracilis, Aphaenogaster picea, Aprostocetus sp?). Which is correct?

Here, we are referring to events. For instance, the three *Leptopilina* gained their 13 EVEs from the same Event and our pipeline correctly counted them as a single event. Similarly, a single event is counted for *M.demolitor* and *C.*vestalis, since they share the same bracovirus deriving from an ancestral nudiviral*.* Finally, our pipeline detected an additional event involving a LbFV-like donor in *C. vestalis* (involving more than 4 genes). In total, this leads to a total of 12 independent events.

Supplementary file 2 – when this file is opened in excel, the column headings are shifted so that they are not present in the correct column. Please define the headings in a separate tab or document – currently, it is very difficult to interpret the information within. Which column indicates domestication (or not)?Figure 2:– In 2A, should an average/median "+" sign be present in the ssRNA column?

It is the case, right in the >260 line. We increased the size of the “+” in order to give more visibility to this sign.

– In 2B, should IVSPERs be in orange?

Orange corresponds to domestication. Our pipeline necessitated either homologous sequences or RNAseq data to infer domestication. Since no such data were available for the Campopleginae species in which the majority of IVSPERs have been found, we were not able to infer domestication and thus, these events should not be coloured in orange, but in yellow.

– 2C, the numbers indicated don't make sense based on other results. Are the EVE numbers inclusive of dEVE numbers? Earlier in the results, it was stated that there are 12 events involving >4 genes for dsDNA viruses. Is this figure depicting events or genes? Line 127 implies genes, and the figure legends and titles imply events.

We agree that the legend in figure 2 did not make sense. Sorry for that. We changed it to “Domestication inferred or not inferred” instead of EVEs and dEVEs and modified accordingly the figure legend.

– The reference to the "excel tab" information needs to be more explicit

We changed it as the Supplementary file 5.

Line 141 and 144: Is there conflicting information presented here about ssRNA viruses being "under-represented" and then "over-represented"?

ssRNA viruses (+ and -) are overall under-represented in endogenization events, but within events involving ssRNA viruses, ssRNA(-) viruses were over-represented: while 32.6% of the RNA viruses in public databases are negative stranded, we found that 72.2% of the events involving RNA viruses implied negative-stranded RNA viruses (and this highly significant).

Where is the data presented to support these statements?

All virus-host associations as well as chi2 test tables can be found in the Supplementary file 5.

Figure 3: The colour in the legend is different from the colour in the figure for ectoparasitioids,

We have changed it.

Figure 5: Where are the methods that were used to make this figure? It is surprising that 142 genes were used to make the alignment. What is the percentage of missing data for each taxon?

Indeed, sorry for this missing point. We added a section called: “*Naldaviricetes* phylogenetic inference “ in the material an method part for that purpose. (Line 707-715).

Here is its content:

“In order to infer the phylogeny of *Naldaviricetes,* we retrieved all predicted ORFs from a set of 25 dsDNA viruses including *Baculoviridae*, *Nudiviridae*, *Hytrosaviridae*, the LbFV-like and AmFV-like viruses, and *Nimaviridae* used as outgroup. We then merged these ORFs with all the candidate EVEs extracted from Hymenoptera genomes and performed an all versus all blastp analysis using Mmseqs2 (Steinegger et al.., 2017). All sequences sharing at least an alignment bit score > 50 and a coverage > 15% were then clustered together. We kept only clusters with more than 3 species (n=142 clusters). Each cluster (or partition) were then aligned using clustalo v1.2.4 (Sievers et al., 2011) and trimmed using Trimal v1.2 (-automated1) (Capella-gutierrez et al.., 2009). Each partition was then concatenated to form one unique protein alignment using the catfasta2phyml.pl script. The *Naldaviricetes* phylogeny was then inferred using Iqtree2 (Minh et al., 2020) using the best model for each partition (Alyaanamoorthy et al.., 2017) (-m MFP -alrt 1000 -bb 1000).”

Line 242: After describing 139 convincing EVEs from P. orseoliae, then 44/89 are mentioned as presenting signs of domestication. Where does the denominator of 89 come from?

Eighty-nine stands for the number of ORFs in the PoFV genome that has homologs in wasp genomes. We tried to clarify by changing the text in ‘(deriving from 89 of the 136 ORFS found in PoFV)’ (lines 279-282).

Line 255: Does AmFV have RNA polymerase subunits or just a single gene encoding the RNA polymerase? If the former, which subunit does AmFV_0047 encode?

AmFV_0047 corresponds to the protein YP_009165796 and according to our Hmmer search, it has an RNA polymerase Rpb1, domain 2 (supplementary file 2). We changed it in the text to clarify (line 300).

Line 275: "we will first discuss"… "a hypothesis for" should be inserted here.

We have changed it.

Line 510: "transportable elements" needs to be fixed.

We have changed it.

Reviewer #2 (Recommendations for the authors):Line 117 – 118 – this sentence is unclear, please revise.

To clarify, we changed the sentence as: “Among the 124 species under study, 113 underwent at least one endogenization event, with a maximum of 14 events and a median of 3” (line 124).

Figure 2 – I would advise adding grid lines to the plots. Now it is very difficult to see how high the bars go for certain lines (I had to use my ruler).

We added more grid lines.

Line 152 – Looking at Figure 3A, it surprises me that ssDNA viruses are not underrepresented in endoparasitoid wasps. The observed percentage is half of the expected percentage.

This is explained by the low sample size for ssDNA (n=25) compared to other genomic structures (for dsDNA n=129).

Line 215 – 216 – is a gene-loss scenario not equally likely?

If we take into account all available genomic data (Campopleginae, Ophioninae, Cremastinae, Ctenopelmatinae, Mesochorinae and Banchinae), one unique ancestral event would have to be lost at least 4 times, which gives 5 events and is less parsimonious than 2 independent acquisitions events (see Figure 1—figure supplement 5).

To emphasize this point, we changed the text to : “Together with previous studies that did not reveal the presence of IVSPER in other internal clades, this result argues against the view of a single event at the root of Ophioniformes, and thus supports the alternative view (Burke et al., 2021) that the so-called IVSPER genes in the Campopleginae and Banchinae subfamilies stem from independent events, despite their striking structural similarities” (Lines 244-245).

Figure 5 – what do the question marks exactly mean here?

We remove the question mark in AprostocetusNV because you are right, the scaffolds of this assembly were unequivocally assigned to this species. However, we kept the “?” for

DoEFV/DoFV because there in uncertainty regarding the endogenous/exogenous status of this sequence. We also added a sentence to explain this question mark in the Figure5.

Line 241 – this was on scaffolds of type A. I would specify here to avoid confusion.

This was for all the scaffolds from A to D (but there was not D score) (see Figure 5supplementary 2-A).

Line 262 – 263 – I am not sure how an absence of raw reads is possible.

Unfortunately, the only genome of *Aphaenogaster picea* available at the time of our analysis was:

GCA_003063865.1 for which we did not find any raw data.

Line 312 "at early times" please clarify.

We changed it as: “Although nuclear replication occurs only at early time post-infection» (line 363).

We agree, and removed this part.

Line 331 – 336 – I would place this section in supplemental.

Since the quality of genomes assemblies varies according to lifestyles, we do think that this should be briefly discussed in the discussion session.

Line 341 – 352 – This section recapitulates to a large extent content that was already outlined in the introduction. For sake of brevity, I would remove it.Reviewer #3 (Recommendations for the authors):In line with my public comment on the scoring system, I feel a more thorough investigation of its impact on the results and conclusions should be conducted such as:– Repeating key conclusions on endogenization/domestication bias for example in being over-inclusive (A to F).

Adding A to F:

Following the reviewer comment, we re-ran the analysis using A to F EVE candidates. The results were overall highly consistent with the previous analysis including only A,B,C and D scores. In particular, we do find the same main result as before: the number of Events and dEvents involving dsDNA viruses is significantly higher in endoparasitoids compared to freeliving species. This conclusion was reached with the whole dataset, and also when we focused on dsDNA viruses (with or without including the controls).

In this respect, we added a few lines at the end of the Material and Method section “Test of the lifestyle effect on viral endogenization and domestication” to discuss the impact of this scoring system on our conclusions. We also complemented the violin plots Figure 4—figure supplement 1 which now contains the GLM coefficients distribution of the model run using scaffolds scored as A only, A to D, and A to F. Furthermore, we also included all the detailed output of the GLM within the supplementary file 6.

Because it was not clear to us whether the reviewer meant to also include X scaffolds, we did this very relaxed analysis. Results were globally consistent with previous analysis in particular for dsDNA viruses. However, we choose not to include this analysis in the manuscript since numerous scaffolds belonging to free-living viruses are inevitably included in this category.

“In our main analysis, we only considered EVEs with scores from A to D as confidently endogenized. To test the impact of this scoring system on our conclusions, we ran the very same analysis on either top-confidence EVEs (A score) (Figure 4—figure supplement 1), or on a more relaxed dataset also including poorly-scored EVEs (A-F). The main conclusions of the manuscript, in particular regarding the link between endoparasitoidism and dsDNA virus endogenization and domestication, were consistently reached (see all statistical summaries of the Bayesian GLM models in Supplementary file 6).” (Lines 781-786).

– Giving a fair representation of the score distribution among different virus genome type.

To give a fair representation of the score distribution among different virus genome type we made a barplot distribution (Figure 1—figure supplement 8).

Additionally, we also added few lines in the Material & Method section “Inference of endogenization event” describing the distribution of scores among virus genomic structures as follow:

Lines 612-617

“The majority of EVEs was found within scaffolds considered as endogenized (A-D score) (Figure 1—figure supplement 8-B). Besides, the distribution of EVEs among scores was roughly identical for each viral genomic structures (Figure 1—figure supplement 8). The main difference was in the fraction of EVEs annotated as X or F. This fraction was higher for DNA viruses compared to RNA viruses (Figure 1—figure supplement 8 -B). Because these scaffolds likely belong to free-living viruses that have been sequenced together with the insect DNA, it is not a surprise to observe an excess of DNA viruses, since DNA and not RNA was sequenced.

– Sharing complete S2 Table (not truncated to the D score).

We added a new excel sheet within the supplementary file 4 named ‘Exogenous viral elements (E-X)” with all the loci scored from E to X.

This would give a better representation of the results. Being over-inclusive has its risks here, and I would be more trusting of the authors' original results, but it might also explain some of the bias (maybe more ssRNA-derived EVE are low scoring, which could explain the bias observed?)

As can be observed in Figure 1—figure supplement 8, ssRNA-derived EVE are not particularly low scoring.

Additional changes

– While looking into the Supplementary file 5, we realized that one of our analyses was possibly biased. This concerned the “Double-stranded DNA viruses are over-represented in endogenization events” section of the paper. We initially used virushostdb complemented with two recent papers (Wu et al., 2020 and Shi et al., 2016) to estimate the virus distribution in insects, in particular according to their genomic structure. However, the two additional papers used were only focusing on RNA viruses, thus biasing the analysis. Therefore, we corrected this mistake by only including data from the virushostdb. The conclusions remained unchanged: dsDNA viruses were much more endogenized compared to their relative abundance in insect species, while ssRNA viruses were under-represented. We took this opportunity to also test whether the distribution of viruses in Hymenoptera was similar to that found in the whole insect class. This was the case (chi^2 = 6.4, df=3, pvalue=0.09). We changed accordingly the values in the main manuscript (lines 147-152), modified the material and methods (lines 694-706), and the supplementary files 5 and 6.

– In line 467 we used to indicate the smallest EVE size found in the controls instead of indicating the average size which is much more informative compared to the minimum N50 value. Therefore, we changed as : “Although some genomes were highly fragmented (such as the 32 genomes we generated since they were obtained using short reads only), the N50 values (min: 3542bp) were equal to or larger than the expected sizes of genes known to be endogenized and domesticated (average size = 1244,4 bp (sd=1105bp)) indicating that most of the putative EVEs should be detected entirely.”

– We added two supplementary source data files including all clusters alignments for two analysis : Figure 1–source data 1 including all aligned cluster sequences scored from A to X and Figure 5–source data 1 including all aligned cluster sequences included in the *Naldaviricetes* phylogenetic analysis.